# Quantitative live cell imaging of a tauopathy model enables the identification of a polypharmacological drug candidate that restores physiological microtubule interaction

Luca Pinzi[1,9], Christian Conze[2,9], Nicolo Bisi[2], Gabriele Dalla Torre[1,7], Ahmed Soliman[2], Nanci Monteiro-Abreu[2], Nataliya I. Trushina[2], Andrea Krusenbaum[2], Maryam Khodaei Dolouei[2], Andrea Hellwig[3], Michael S. Christodoulou[1,4,8], Daniele Passarella[4], Lidia Bakota[2], Giulio Rastelli[1] ✉ & Roland Brandt[2,5,6] ✉

Tauopathies such as Alzheimer's disease are characterized by aggregation and increased phosphorylation of the microtubule-associated protein tau. Tau's pathological changes are closely linked to neurodegeneration, making tau a prime candidate for intervention. We developed an approach to monitor pathological changes of aggregation-prone human tau in living neurons. We identified 2-phenyloxazole (PHOX) derivatives as putative poly-pharmacological small molecules that interact with tau and modulate tau kinases. We found that PHOX15 inhibits tau aggregation, restores tau's physiological microtubule interaction, and reduces tau phosphorylation at disease-relevant sites. Molecular dynamics simulations highlight cryptic channel-like pockets crossing tau protofilaments and suggest that PHOX15 binding reduces the protofilament's ability to adopt a PHF-like conformation by modifying a key glycine triad. Our data demonstrate that live-cell imaging of a tauopathy model enables screening of compounds that modulate tau-microtubule interaction and allows identification of a promising poly-pharmacological drug candidate that simultaneously inhibits tau aggregation and reduces tau phosphorylation.

Tauopathies are a group of neurodegenerative disorders associated with the accumulation of abnormal tau protein in the brain[1]. The most common tauopathy is Alzheimer's disease (AD), in which intracellular aggregates of tau with increased phosphorylation (hyperphosphorylation) are joined by extracellular amyloid plaques containing aggregated amyloid-β (Aβ)[2]. Because tau mutations are sufficient to cause tauopathies and because tau inclusions correlate much better with cognitive impairment than amyloid plaques do, tau pathology is considered to be the major driver for neuronal degeneration in AD and other tauopathies[3–6].

Due to the failure of many Aβ-targeted therapies, tau has become a target of rapidly evolving therapeutic strategies[7,8]. However, tau is a

challenging target due to its complex interactions as an intrinsically disordered protein, its various post-translational modifications, and the complexity of tau pathologies[9–11]. Mechanisms of toxicity are also controversial, but in addition to larger tau fibrils, soluble tau oligomers and tau with disease-like modifications can be toxic species[12–14]. Furthermore, disruption of the physiological function of tau, particularly its dynamic interaction with axonal microtubules, can be a crucial contributor to the disease[15].

Because abnormal forms of tau could trigger a plethora of pathomechanisms, targeting individual downstream mechanisms may have limited therapeutic effects[7,16]. Polypharmacological drugs may provide a solution to this problem by simultaneously modulating tau aggregation, tau phosphorylation, and potential additional tau functions that may be impaired due to disease. In fact, polypharmacological approaches have shown significant advantages over single-target drugs, particularly in the treatment of complex diseases[17,18]. Cell-based models of tauopathies that would help to monitor tau activity and behavior in neurons and to develop mechanism-based therapies are scarce. Previously, biosensor cells to monitor tau aggregation have been generated[19], however, the cell lines are based on the expression of artificial tau fragments, and the relevance for the formation of authentic amyloidogenic tau aggregates has been questioned[20]. In addition, since the tau regions involved in aggregation and microtubule-binding overlap, it would be important to determine whether tau aggregation inhibitors that bind covalently or non-covalently to the microtubule-binding region of tau[21] interfere with the physiological interaction of tau with microtubules.

In this work, we developed a live-cell imaging assay to identify compounds that restore physiological microtubule interaction of an aggregation-prone full-length human tau construct in axon-like processes of model neurons and axons of primary neurons. Using a panel of small molecules predicted to have tau and kinase modulating

activity, we identified the 2-phenyloxazole derivative PHOX15, which restores the physiological tau-microtubule interaction in cells. We show that this small molecule inhibits tau aggregation in vitro, reduces formation of tau amyloids in primary neurons, inhibits the tau kinases GSK3β and Cdk5, and decreases tau phosphorylation at selected pathogenic sites in the proline-rich region of tau. Molecular dynamics (MD) simulations identified cryptic channel-like pockets crossing tau protofilaments not previously described. Extended MD simulations conducted on different conformations and aggregation states of tau suggested that binding of PHOX15 to the protein could significantly decrease the protofilament's ability to adopt a paired helical filament (PHF)-like conformation, consistent with that PHOX15 exerts an inhibitory effect on tau aggregation.

## Results

### Aggregation-prone human tau exhibits increased aggregation in vitro and reduced microtubule interaction in neuronal cells

We used the single amino acid deletion mutant TauΔK280 reported in two cases of tauopathies[22,23] as a tool to develop a cellular assay to test for compounds affecting tau aggregation and improving the tau-microtubule interaction. In vitro experiments had previously demonstrated that this mutant showed an increased propensity to form paired helical filaments[24]. Consistent with this, recombinant human TauΔK280 showed greater than 50% increased aggregation compared to wild-type tau in heparin-induced cell-free aggregation assays (Fig. 1A). In standard in vitro microtubule-polymerization assays, wild-type tau and TauΔK280 showed very similar activity in promoting microtubule assembly, consistent with little or no tau aggregation under cell-free conditions in the absence of heparin.

We used fluorescence decay after photoactivation (FDAP) experiments to determine a possible change in the interaction of TauΔK280 with microtubules in axon-like processes of model neurons.

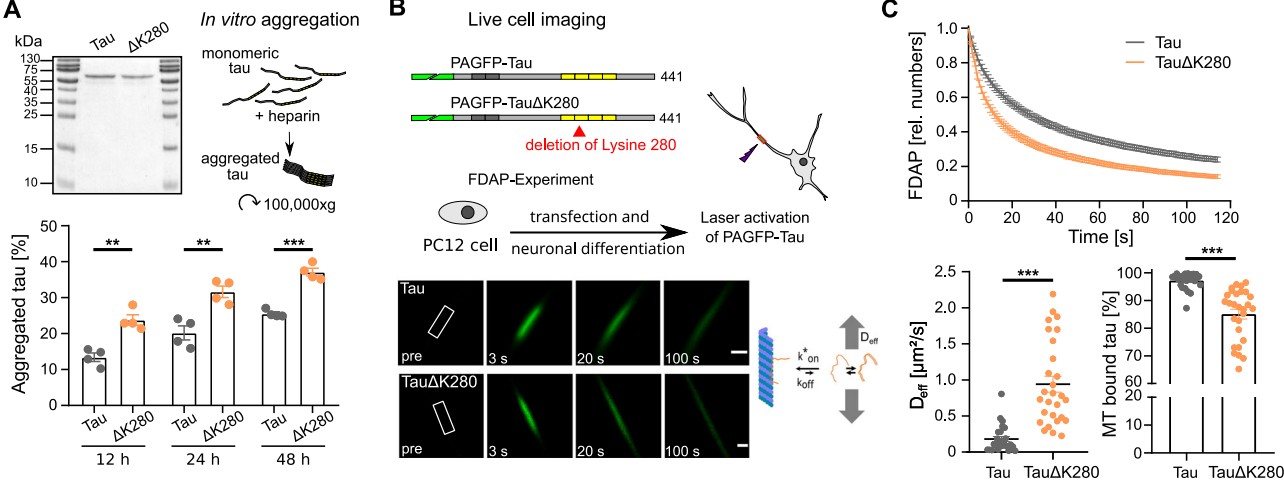

**Fig. 1 | Aggregation-prone human tau exhibits increased aggregation in vitro and reduced microtubule interaction in neuronal cells. A** In vitro aggregation assays show increased aggregation of recombinant aggregation-prone human tau (TauΔK280) compared to wild-type tau (Tau). Coomassie Brilliant Blue-stained SDS-PAGE of the purified proteins used for the aggregation assays (top left) and fractions of aggregated tau determined by ultracentrifugation after incubation with heparin for the indicated times are shown (bottom; mean ± SEM, $n = 4$). Molecular mass standards are indicated. Statistically significant differences between Tau and TauΔK280 as determined by an unpaired two-tailed Student's t-test are displayed. **$p < 0.01$; ***$p < 0.001$ (individual $p$ values: 12 h: 0.0014, 24 h: 0.0040, 48 h: <0.0001). **B** Live cell imaging of tau dynamics in axon-like processes of neuronally differentiated PC12 cells. Schematic representations of the expressed tau constructs are shown above. The MT-binding repeat regions (RR1–RR4) are indicated by yellow boxes, and the N-terminal PAGFP fusion in green. Adult-specific exons in

the N-terminus of tau (N1, N2) are shown in dark gray. The lysine deletion of the aggregation-prone construct is in the second repeat. Shown below are representative time-lapse micrographs of a fluorescence decay after photoactivation (FDAP) experiment. Scale bar, 3 μm. The scheme shows that calculation of the effective diffusion constant ($D_{eff}$) using a one-dimensional diffusion model allows the fraction of microtubule-bound tau to be determined[25]. **C** FDAP diagrams after photoactivation of PAGFP-Tau- or PAGFP-TauΔK280-expressing PC12 cells show increased fluorescence decay of TauΔK280. Mean ± SEM of 26 (Tau) and 28 (TauΔK280) cells are shown. Scatterplots of effective diffusion constants ($D_{eff}$) and percentage of tau bound to microtubules are shown below indicating decreased microtubule binding of TauΔK280. Statistically significant differences between Tau and TauΔK280 as determined by unpaired two-tailed Student's t-tests with Welch correction are indicated. ***$p < 0.001$ ($D_{eff}$: <0.0001, MT bound tau: <0.001). Source data are provided as a Source Data file.

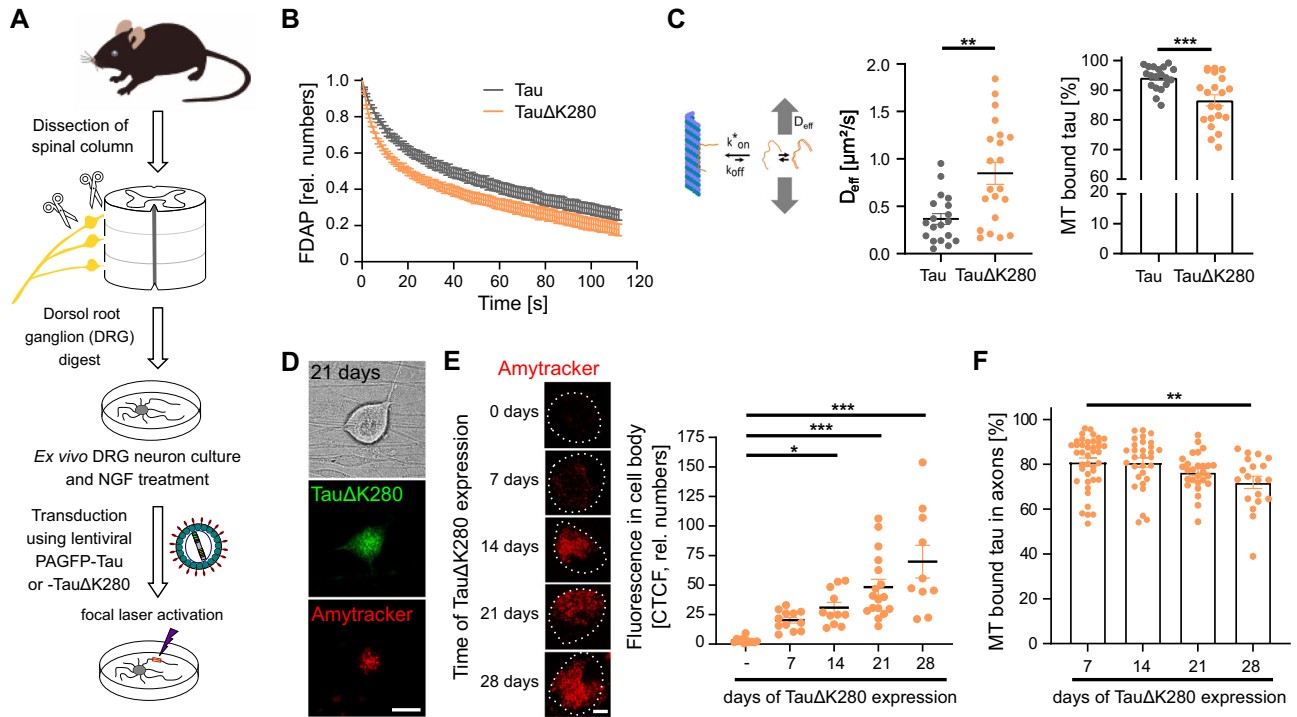

**Fig. 2 | Long-term expression of aggregation-prone tau leads to the progressive formation of tau amyloids in primary neurons. A** Preparation of primary neurons from dorsal root ganglia (DRG). DRG neurons were prepared from adult mice by extracting the ganglia delineating the spinal cord and sequential enzymatic digestion. Neurons were lentivirally transduced for long-term expression of PAGFP-Tau or PAGFP-TauΔK280 and subjected to FDAP experiments. **B** FDAP diagrams after photoactivation of axonal segments in PAGFP-Tau or PAGFP-TauΔK280-expressing DRG neurons show increased fluorescence decay of TauΔK280 also in primary neurons. Mean ± SEM of 19 (Tau) and 22 (TauΔK280) cells are shown. **C** Scatterplots of effective diffusion constants ($D_{eff}$) (left) and percentage of tau bound to microtubules (right) in axons of DRG neurons as determined by modeling of FDAP diagrams indicating decreased microtubule binding of TauΔK280. Mean ± SEM of 19 (Tau) and 21 (TauΔK280) cells (left and right) are shown. Statistically significant difference between Tau and TauΔK280 as determined by unpaired two-tailed Student's t-tests with Welch correction are indicated. **\*\****p* < 0.01, **\*\*\****p* < 0.001 ($D_{eff}$: <0.002, MT bound tau: <0.001). **D** Micrographs of a DRG neuron after 3 weeks of TauΔK280 expression showing a transmitted light micrograph, PAGFP-TauΔK280 fluorescence, and the Amytracker™ 680 signal indicating the formation of tau amyloids in the neuronal cell body. Scale bar, 20 μm. **E** Amytracker™ staining of cell bodies of PAGFP-TauΔK280-expressing DRG neurons over time indicate progressive formation of tau amyloids. Quantification of fluorescence in the cell body by calculation of corrected total cell fluorescence (CTCF) indicates progressive formation of tau amyloids over time. Shown are mean ± SEM of CTCF values from 10-17 TauΔK280-expressing cells (d0: 11, d7: 12, d14: 11, d21: 17, d28: 10). Statistically significant differences from non-transduced cells (-) determined by one-way ANOVA with Dunnett's post hoc test are indicated. \**p* < 0.05, \*\*\**p* < 0.001 (- vs. 14: 0.029, - vs. 21: <0.001, - vs. 28: <0.001). Scale bar, 20 μm. **F** Long-term TauΔK280-expression leads to decreased tau-microtubule interaction in axons of DRG neurons as measured by FDAP experiments. Shown are scatterplots of the percentage of tau bound to microtubules as determined by modeling the FDAP diagrams after photoactivation of axonal segments. Mean ± SEM of 19-41 (d7: 41, d14: 30, d21: 27, d28: 19) neurons is shown. Statistically significant differences from the 7-day value determined by one-way ANOVA with Dunnett's post hoc test are indicated. \*\**p* < 0.01 (28 vs. 7: 0.0092). Source data are provided as a Source Data file.

---

We have previously shown that microtubule interaction reduces the effective diffusion of tau by a factor of 10 as measured by FDAP experiments, and that changes in the effective diffusion constant ($D_{eff}$) can be used to quantify the interaction of tau with microtubules in living neuronal cells[15,25,26]. The tau constructs were N-terminally tagged with photoactivatable GFP (PAGFP) to minimize potential interference with tau-microtubule and tau-tau interaction and were exogenously expressed in PC12 cells, which were differentiated into a neuronal phenotype. After focal photoactivation, both constructs showed dissipation from the region of photoactivation that was higher for TauΔK280 (Fig. 1B). Indeed, the FDAP curves showed increased decay of TauΔK280 compared to wild-type tau, indicating reduced interaction with microtubules in the cellular environment (Fig. 1C, top). Accordingly, calculation of the effective diffusion constant using a previously established one-dimensional diffusion model[26] resulted in a significantly higher value for TauΔK280. This corresponds to a > 10% reduced binding to microtubules of TauΔK280 (Fig. 1C, bottom), likely caused by the formation of soluble tau oligomers in the cells. Of note, the tau oligomers were small enough not to cause reduced diffusion in our assay system. This was consistent with our observation that they

were negative for staining with the optotracer Amytracker™ (Ebba Biotech, Sweden), which detects higher molecular weight tau amyloids.

## Long-term expression of aggregation-prone tau leads to the progressive formation of tau amyloids in primary neurons

To determine whether TauΔK280 exhibits decreased microtubule interaction as a result of oligomer formation in authentic neuronal axons as well, we transduced primary dorsal root ganglia (DRG) neurons dissected from adult mice to express the respective PAGFP-tagged tau constructs (Fig. 2A). Again, TauΔK280 showed higher effective diffusion and a very similar decrease in axonal microtubule-binding of TauΔK280 compared to wild-type-tau, consistent with the formation of soluble TauΔK280 oligomers also in primary neurons (Fig. 2B, C). To test the hypothesis that the tau oligomers are precursors for the formation of larger, insoluble tau aggregates, we analyzed DRG neurons 3 weeks after lentiviral transduction to determine the long-term effect of TauΔK280 expression in the cells. Indeed, at this time point, tau aggregates could be observed in most cell bodies of TauΔK280 expressing cells (Fig. 2D). The tau aggregates were

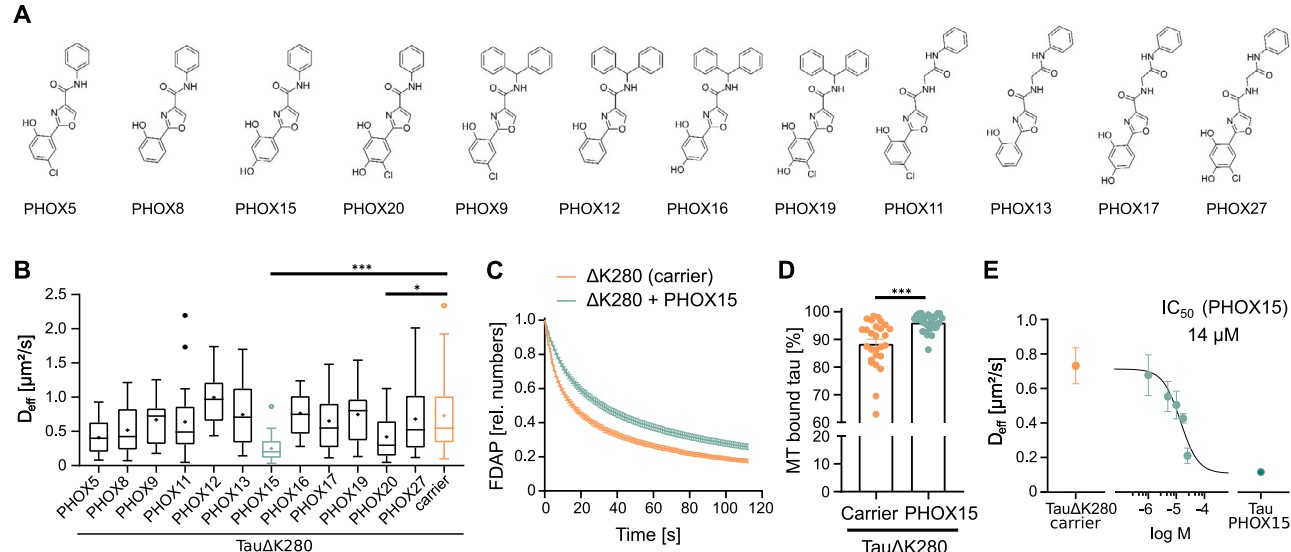

**Fig. 3 | PHOX15 restores the physiological microtubule interaction of aggregation-prone tau. A** Chemical structures of the investigated PHOX compounds. **B** Boxplots of effective diffusion constants ($D_{eff}$) in response to the twelve PHOX derivatives shown in (**A**) identifies two compounds (PHOX15, PHOX20) inducing decreased effective diffusion of TauΔK280. TauΔK280-expressing neuronally differentiated PC12 cells were treated with either 25 μM compound or carrier (0.125% DMSO) for 20 hours prior to imaging. Tukey boxplot with median (·) and mean (+) of 15-30 cells per condition is shown (PHOX5: 26, PHOX8: 20, PHOX9: 15, PHOX11: 21, PHOX12: 23, PHOX13: 20, PHOX15: 30, PHOX16: 25, PHOX17: 23, PHOX19: 25, PHOX20: 27, PHOX27: 18, carrier: 27). Bottom and top of the boxes show the 25th and 75th percentiles. Whiskers represent the expected variation of the data (1.5 times the interquartile range (IQR)). Outliers are indicated. Statistically significant differences in compound to carrier were determined by one-way ANOVA with Dunnett's post hoc. *$p < 0.05$; ***$p < 0.001$ (PHOX15 vs. carrier: <0.0001, PHOX20 vs. carrier: 0.0454). **C** FDAP plots after photoactivation of PAGFP-

positive after live staining with Amytracker™ (Ebba Biotech, Sweden[27], indicating formation of tau amyloid with more than eight parallel β-sheets in register. To determine the progression of tau amyloid formation over time, we performed Amytracker™ staining at different time points after infection. Quantification of fluorescence in the cell body revealed a constant increase in corrected total cell fluorescence (CTCF), indicating progressive formation of tau aggregates over time (Fig. 2E). The increase in fluorescence became significant after 2 weeks of TauΔK280 expression. Parallel to the formation of aggregates in the cell body, the proportion of TauΔK280 binding to axonal microtubules progressively decreased with expression time (Fig. 2F). This indicates that long-term expression of TauΔK280 in neurons leads to increased formation of soluble tau oligomers, leading to formation of tau amyloids in the cell body.

Taken together, the results demonstrate a reduced microtubule interaction of aggregation-prone TauΔK280 in axon-like processes of model neurons and axons of primary neurons. The time course of TauΔK280-expression in primary neurons indicates that the reduced microtubule interaction is caused by the formation of small soluble tau oligomers as precursors to tau amyloids that form after prolonged tau exposure. The data also point out that this imaging approach provides a useful tool to identify compounds that modulate tau oligomerization and tau-microtubule interaction in axons of living neurons.

### Chemoinformatic analyses indicate that PHOX compounds can inhibit tau aggregation

We have previously identified 2-phenyloxazole (PHOX) derivatives as selective monoamine oxidase B (MAO-B) inhibitors[28] (Fig. 3A). MAO-B inhibitors are used to treat Parkinson's disease and are also considered

TauΔK280-expressing PC12 cells with the most active compound (PHOX15) or carrier (0.125% DMSO). Mean ± SEM of 30 (PHOX15) and 27 (carrier) cells are shown. **D** Scatterplot of percentage of tau bound to microtubules as determined by modeling of FDAP diagrams indicates that PHOX15 induces increased binding of TauΔK280 to microtubules. Mean ± SEM of 30 (PHOX15) and 27 (carrier) cells are shown. Statistically significant difference determined by unpaired two-tailed Student's t-tests with Welch correction is indicated. ***$p < 0.001$ (<0.0001). **E** Dose-response curve of the effect of PHOX15 on the effective diffusion constants ($D_{eff}$) of TauΔK280-expressing PC12 cells indicates an $IC_{50}$ of 14 μM. Means ± SEM of 15-19 cells per concentration (1 μM: 15, 5 μM: 16, 10 μM: 18, 17.5 μM: 19, 25 μM: 15 cells) are shown. The curve was fitted using a four-parameter log-logistic model. The plateau of the curve was constrained to $D_{eff}$ of TauΔK280 with carrier (0.125% DMSO, $n = 27$) and Tau with 25 μM PHOX15 ($n = 26$) according to[75]. Source data are provided as a Source Data file.

potential drug candidates for the treatment of AD[29,30]. The ability of compounds to simultaneously modulate multiple pathways involved in the pathogenesis and progression of complex diseases is of central interest. In accordance with the concept of polypharmacology, i.e., the use of a single drug acting on multiple targets or disease pathways[17], we searched for possible additional targets of the PHOX compounds through an integrated 2D (MACCS and ECFP4 fingerprints) and 3D computational similarity approach[31], which allowed us to prioritize a set of potential targets of the PHOX compounds according to their similarity with active ChEMBL ligands[32] (Table S1). Remarkably, the microtubule-associated protein tau was one of the best-scoring targets, with compounds emerging from consensus of 2D and 3D similarity methods. Other targets emerged from 2D similarity estimations only, including Glycogen Synthase Kinase 3β (GSK3β)[33], or from 3D similarities. Representative 3D alignments with high conformational overlap and significant superposition of hydrogen-bond acceptor/donor and aromatic features are reported in Fig. S1.

We further investigated whether PHOX compounds might possess molecular properties and scaffolds required for tau anti-aggregation activity[34]. The vast majority of the PHOX compounds met these requirements (Fig. S2 and Table S3), with the exception of the bulkier benzylbenzene derivatives PHOX 9, 12, 16, and 19. In particular, the molecular properties of PHOX 5, 8, 11, 13, and 15 were similar to those of highly active tau ligands (Fig. S2 and Table S4). These compounds have an amide group, three aromatic rings (one of which being a heterocycle), around six hydrogen-bond acceptors and a low aliphatic character[34]. In addition, the compounds were tested in silico for their predicted ability to cross the blood-brain barrier by using the QikProp software (Schrödinger Release 2022-1: QikProp;

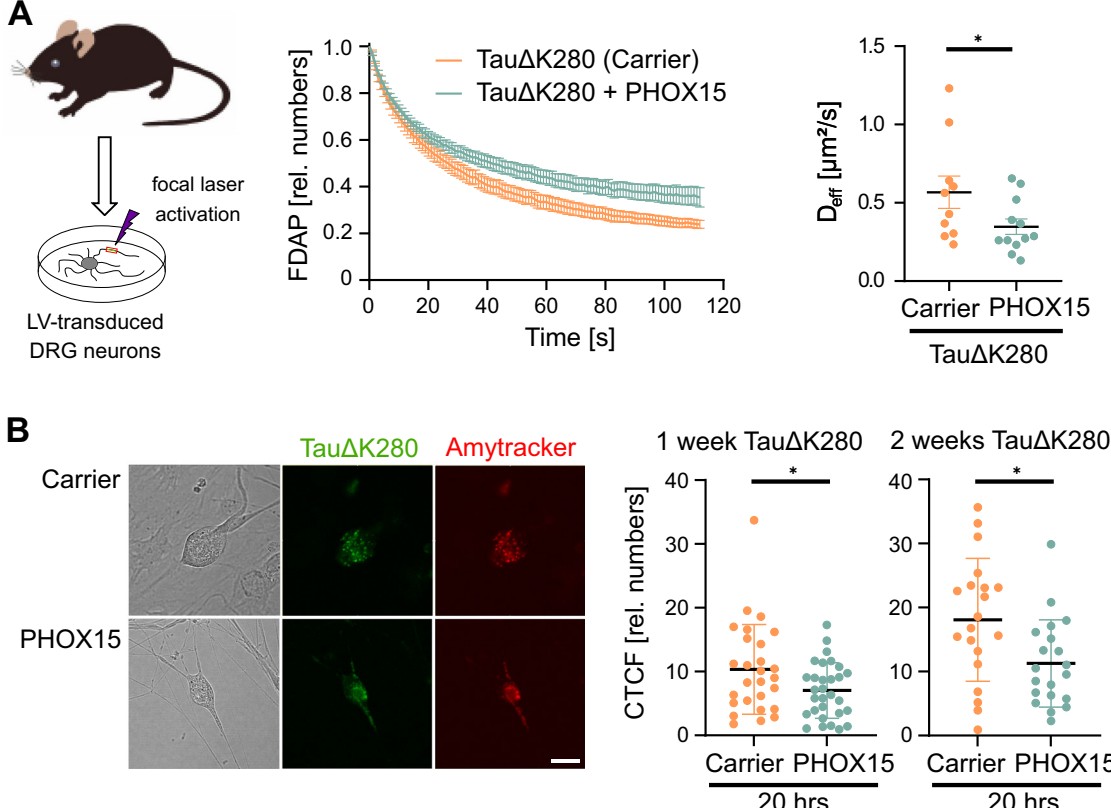

**Fig. 4 | PHOX15 reduces the formation of tau amyloids in neurons. A** FDAP plot and scatterplot of effective diffusion constants ($D_{eff}$) after photoactivation of lentivirally transduced DRG neurons expressing PAGFP-TauΔK280 shows that PHOX15 decreases effective diffusion of aggregation-prone tau also in axons of primary neurons. The effect of PHOX15 (25 μM, treatment of 20 hours prior to imaging) compared to a carrier control (0.125% DMSO) is shown. Mean ± SEM from 12 (PHOX15) and 10 cells (control) are indicated. Statistically significant differences between samples determined by an unpaired two-tailed Student's t-test are indicated. *$p < 0.05$ ($p = 0.0495$). **B** PHOX15 treatment decreases the amount of tau amyloids in PAGFP-TauΔK280-transduced cells as determined by Amytracker™ staining. Scatterplots of corrected total cell fluorescence (CTCF) of the cell bodies of DRG neurons after 1 and 2 weeks of TauΔK280 expression treated for 20 hours with 25 μM PHOX15 or carrier (0.125% DMSO) are shown on the right. Shown are mean ± SEM of CTCF values from 30 (PHOX15) and 27 cells (control) after 1 week, and 20 (PHOX15) and 20 cells (carrier) after 2 weeks of TauΔK280-expressing cells. Statistically significant difference determined by unpaired two-tailed Student's t-tests is indicated. *$p < 0.05$ (1 week: 0.0347, 2 weeks: 0.0136). Representative micrographs showing transmitted light images, PAGFP-TauΔK280 fluorescence, and Amytracker™ 680 signal of DRG neurons after 2 weeks of TauΔK280 expression and treatment with carrier or PHOX15 for 20 hrs are shown on the left. Scale bar, 20 μm. Source data are provided as a Source Data file.

Schrödinger, LLC: New York, NY, USA, 2022) (Table S5), resulting in permeability values commonly observed for drugs acting on the central nervous system (CNS).

Taken together, the results of the chemoinformatic analyses suggest that the PHOX compounds affect tau oligomerization and can potentially cross the blood-brain barrier, two important features that make these compounds worthy of further investigation.

### The 2-phenyloxazole derivative 2-(2,4-dihydroxyphenyl)-N-phenyloxazole-4-carboxamide (PHOX15) restores the physiological microtubule interaction of aggregation-prone tau

The twelve PHOX derivatives shown in Fig. 3A were tested for their effect on metabolic activity and cytotoxicity profiles on differentiated model neurons. While some PHOX derivatives affected MTT conversion in a concentration-dependent manner, indicating biological activity, none of the compounds were toxic at concentrations of 25 μM or below (Fig. S3).

Next, we used our live cell imaging approach to test the PHOX derivatives for their activity to reduce tau oligomerization and increase microtubule interaction of TauΔK280 at subtoxic concentrations. Two of the compounds (PHOX15 and PHOX20) significantly reduced the effective diffusion constant, with PHOX15 being the most efficient (Fig. 3B). Indeed, the FDAP curves showed a significantly reduced

dissipation of TauΔK280 in the presence of PHOX15 compared to a vehicle control (Fig. 3C), indicating increased microtubule interaction. Accordingly, mathematical modeling of the FDAP curves revealed an ~10% increased binding of TauΔK280 to microtubules, almost to the level of wild-type tau protein (Fig. 3D). We also determined the $IC_{50}$ value of the activity of PHOX15 to restore microtubule binding of TauΔK280, obtaining a value of 14 μM, which is in the concentration range of the expressed tau construct (Fig. 3E). The results suggest that PHOX15 binds to aggregation-prone tau, inhibits tau oligomerization, and restores the physiological interaction between tau and microtubules in axon-like processes of neuronal cells.

### PHOX15 reduces tau aggregation in vitro and in cells

Next, we tested whether PHOX15 restores the interaction of TauΔK280 with microtubules also in authentic neurons and reduces the formation of tau amyloids in the cells. As primary neuronal system, we used the DRG neurons transduced to express TauΔK280 as described above (see Fig. 2A). We observed that treatment with PHOX15 also reduced the dissipation of TauΔK280 in authentic axons compared to a vehicle control and consequently significantly reduced the effective diffusion constant (Fig. 4A). This indicates that PHOX15 is able to restore the physiological interaction of aggregation-prone tau with microtubules also in primary neurons.

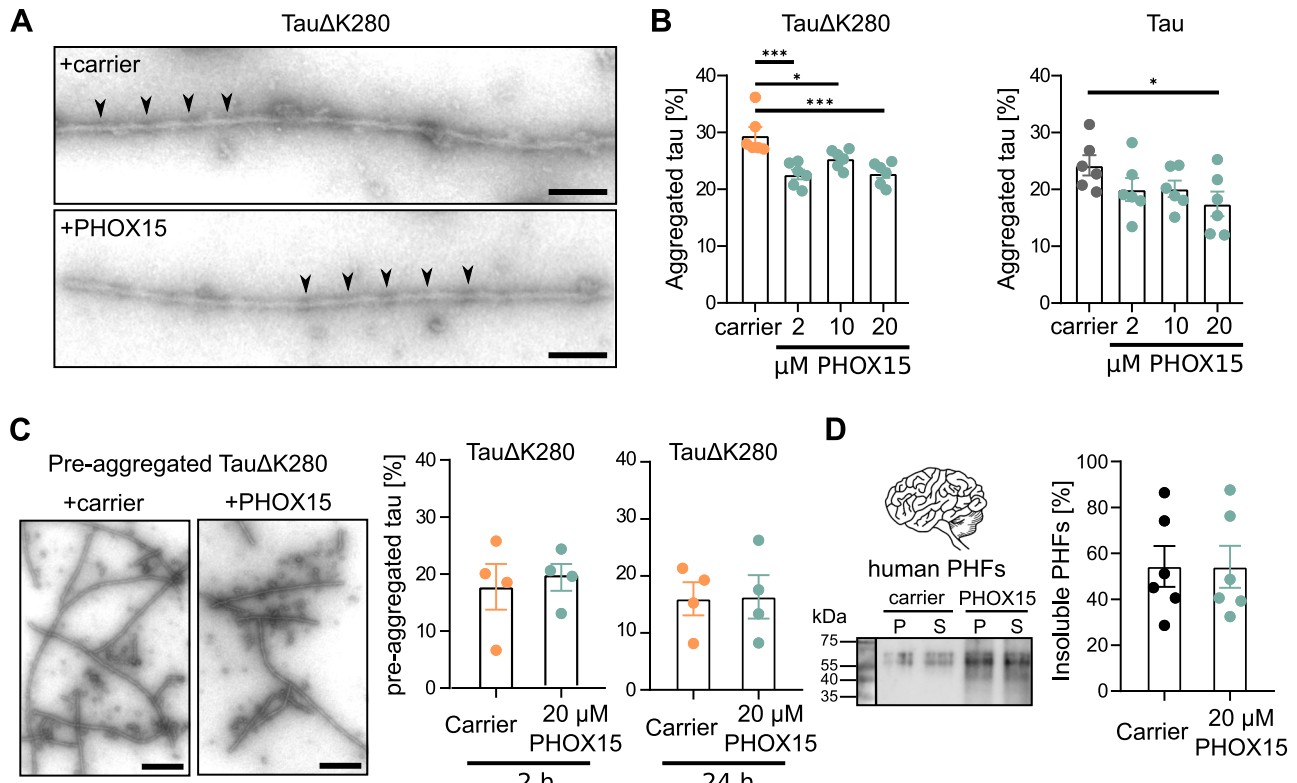

**Fig. 5 | PHOX15 decreases tau aggregation in vitro without affecting existing filaments. A** Electron micrographs of tau filaments that were formed in the presence of heparin from purified TauΔK280 (0.4 mg/ml) in the presence of PHOX15 (20 μM) or carrier (1% DMSO) indicates that PHOX15 does not affect filament morphology. Samples were incubated for 24 hrs at 37 °C, and negatively stained with 1% uranyl acetate. Scale bar, 100 nm. Thin regions of the cylindrical filaments marked by arrowheads indicate a helical turn period of 70-80 nm. **B** PHOX15 decreases aggregation of recombinant TauΔK280 and human wild-type tau (Tau) in vitro. Fractions of aggregated tau determined by ultracentrifugation after incubation for 24 hrs with the indicated concentration of PHOX15 or carrier (1% DMSO) are shown (mean ± SEM; $n = 6$). Statistically significant differences between PHOX15-treated samples and control as determined by one-way ANOVA followed by a Dunnett's post hoc test. *$p < 0.05$; ***$p < 0.001$ (TauΔK280: 2 vs. carrier:

0.0003, 10 vs. carrier: 0.0235, 20 vs. carrier: 0.0004, Tau: 20 vs. carrier: 0.0494). **C** PHOX15 does not affect the morphology and amount of pre-aggregated filaments. Electron micrographs of tau filaments (24 hrs pre-aggregated TauΔK280) treated with PHOX15 (20 μM) or carrier (1% DMSO) for additional 24 hrs. Fractions of aggregated tau determined by ultracentrifugation are shown on the right (mean ± SEM; $n = 4$). Scale bar, 250 nm. **D** PHOX15 does not disaggregate PHFs from human brain. Samples of 0.1 mg/ml SDS-soluble PHFs[36] were treated for 24 hrs with PHOX15 (20 μM) or carrier (1% DMSO). Immunoblot of the pellet (P) and (S) supernatant fractions of the PHFs stained with PHF1 antibody[71] (left panel). Molecular mass standards are indicated. Fractions of insoluble PHFs determined by ultracentrifugation are shown on the right (mean ± SEM; $n = 6$). Source data are provided as a Source Data file.

To determine whether PHOX15 reduces tau aggregation in the cells, we transduced DRG neurons for long-term expression of TauΔK280 to induce the formation of Amytracker™-positive tau amyloids as described above (see Fig. 2E). One or two weeks after transduction, cells were treated with PHOX15 or a vehicle control for 20 hours and stained with Amytracker™. We observed Amytracker™-positive tau aggregates in most neuronal cell bodies under all conditions (Fig. 4B, left). However, quantification of fluorescence showed significantly fewer tau aggregates in PHOX15-treated cells compared to vehicle control, indicating that PHOX15 inhibits tau amyloid formation at both time points (Fig. 4B, right). Note that filaments progressively formed after one and two weeks in both the presence and absence of PHOX15, suggesting that PHOX15 inhibits filament formation but does not dissolve tau amyloids once they have formed in the cell.

To determine whether PHOX15 affects tau aggregation directly and whether it affects existing tau filaments, we turned to our heparin-induced cell-free aggregation assays using wild-type recombinant tau and TauΔK280 (see Fig. 1A). Recombinant TauΔK280 assembled into single filament structures with a common helical turn period very similar to what was previously observed for cylindrical filaments derived from PHFs[35]. The filaments had a diameter of 9-12 nm and a helical turn period (distance between the thin regions) of 70-80 nm (Fig. 5A). The presence of PHOX15 had no significant effect on the

structure of the filaments, and the diameter of the cylindrical filaments and the turn period were similar. However, PHOX15 reduced filament formation by approximately 20%, as quantified by sedimentation assays (Fig. 5B). When added to preformed tau aggregates, PHOX15 had no effect on the structure and amounts of filaments (Fig. 5C). Consequently, treatment with PHOX15 did not decrease the amount of SDS-soluble PHFs isolated from human brain either[36] (Fig. 5D).

Thus, the results suggest that PHOX15 reduces tau aggregation in both cells and cell-free reactions by directly inhibiting filament formation. Cylindrical filaments with periodically thin regions have been interpreted as PHF precursor filaments[35], suggesting that PHOX15 also reduces the formation of PHFs and NFTs. However, it should be noted that recombinant heparin-induced tau fibrils and tau filaments isolated from patients with tauopathies have striking structural differences[37].

## PHOX15 inhibits the tau kinases GSK3β and Cdk5, changes the kinome activity of the cells, and reduces tau phosphorylation at disease-relevant sites

Aberrant phosphorylation and aggregation are key features of tau in the brains of AD patients, and high stoichiometric increased tau phosphorylation (hyperphosphorylation) is thought to result in tau dysfunction and pathological properties[1,38]. Several kinases have been implicated in the aberrant phosphorylation of tau and are thought to

contribute to disease progression. Of these, glycogen synthase kinase 3β (GSK3β) and cyclin-dependent kinase 5 (Cdk5) have been implicated in AD[33,39].

Notably, GSK3β also emerged as a top-ranked target in the similarity estimates reported in Table S1, further reinforcing the putative polypharmacological behavior of this ligand. To assess whether PHOX15 fits into the tau kinase GSK3β, docking calculations were performed in a selected crystal structure (i.e., PDB code: 4AFJ[40]) using the Induced Fit Docking protocol (IFD) implemented in Maestro (Schrödinger suite 2022-1), as detailed in the Supporting Information. The results of the docking calculations suggest that PHOX15 fits into the adenosine triphosphate (ATP) binding pocket of GSK3β (Fig. 6A). The amide carbonyl of PHOX15 hydrogen bonds with the backbone nitrogen of the Val92 hinge, the phenyl ring occupies the hydrophobic pocket of the kinase lined by Ile24, Val32, Tyr91 and Leu145 residues, while the para-hydroxyl group hydrogen bonds with both the Glu56 residue of the αC helix and the Asp157 residue of the highly conserved Asp-Phe-Gly (DFG) motif. Notably, in vitro kinase assays on these targets showed that PHOX15 inhibited both GSK3β and Cdk5 activity with $IC_{50}$ values of 1.9 and 1 μM, respectively (Fig. 6B), confirming the predictions. Docking of PHOX15 to CDK5 resulted in a binding mode similar to that observed in GSK3β (see Figure D in the Supporting Information).

To systematically determine the effect of PHOX15 on gene expression and protein phosphorylation, we performed proteomics and phosphoproteomics analyses of model neurons treated with PHOX15 or a vehicle (Figs. 6C and S4). None of the >60 experimentally confirmed tau interaction partners[38] was significantly down- or up-regulated in PHOX15-treated neurons (Fig. 6D). This suggests that the changes in tau aggregation and tau-microtubule interaction are independent of the differential expression of components of the tau interactome.

To determine the effect of PHOX15 on the kinome activity of the cells, we performed a kinase enrichment analysis to link the identified phosphorylation sites to the reduced activity of the kinases that are most likely responsible for the decreased protein phosphorylation[41,42]. GSK3β ranked fourth in the list (Fig. 6E), indicating that PHOX15 affected GSK3β-dependent phosphorylation in neural cells as well.

Next, we determined the effect of PHOX15 on the phosphoprofile of endogenous tau by phosphoproteomics. Phosphorylation of PHOX15-treated neurons was greatly reduced at several serine and threonine residues in the proline-rich region (PRR) of tau amino-terminally flanking the microtubule-binding region (Figs. 6F and S5). Tau's PRR is considered to be a signaling module that regulates tau's intracellular interactions, including its activity to interact with microtubules[9]. The PRR has the highest relative content of serine/threonine residues, making it the major region of tau for phosphorylation[9]. Phosphorylation of multiple sites, particularly a phosphocluster between Thr231 and Ser239, showed a large decrease in phosphorylation after PHOX15 treatment. This phosphocluster includes major phosphorylation sites identified in PHFs from patients with AD (Thr231, Ser235)[43], which are predicted phosphorylation sites for GSK3β and Cdk5. Other sites of severe reduction include Thr181, an established core biomarker for cerebrospinal fluid (CSF) tau in AD[44], which is also predicted to be phosphorylated by GSK3β and Cdk5 (NetPhos-3.1b prediction[45]). Notably, Thr231 was recently identified as a master site governing the propagation of tau phosphorylation at several AD-associated epitopes[46].

To determine whether PHOX15 increases microtubule interaction by reducing tau phosphorylation at disease-relevant sites, we performed FDAP assays to assess a possible change in the interaction of wild-type tau and a phosphoblocking tau construct with microtubules in axon-like processes of model neurons. The phosphoblocking tau construct was designed by mutating to alanine the ten major phosphorylation sites previously identified as hyperphosphorylated in tau

from AD patients[38,43], to prevent phosphorylation at these residues. Five of these sites were in the PRR (Ser198, Ser199, Ser202, Thr231, Ser235); all showed reduced phosphorylation as a result of PHOX15 treatment (Fig. 6F). Treatment with PHOX15 increased microtubule association of wild-type tau but failed to do so with phosphoblocking tau (Fig. 6G). This finding suggests that PHOX15 has a dual effect on tau-microtubule interaction. First, it restores the physiological interaction of an aggregation-prone construct by reducing tau aggregation, and second, it increases microtubule interaction by reducing tau phosphorylation at disease-relevant sites.

## Molecular dynamics simulations highlight cryptic channel-like pockets crossing tau protofilaments

Molecular dynamics (MD) simulations were performed starting from two different conformations and aggregation status of tau, i.e., from representatives of both 4R and 4R-3R isoforms available among those reported into the Protein Data Bank[47,48]. As for 4R-3R, we selected the cryo-EM structure of the tau filaments from AD brains (PDB ID: 5O3L)[49,50] while for 4R we considered the cryo-EM structure of tau filaments from Progressive Supranuclear Palsy (PSP) (PDB ID: 7P65)[11].

The 5O3L cryo-EM structure of 4R-3R tau reveals a C-shaped structure composed of stacked tau sequences encompassing residues 306-378 of tau's microtubule-binding domain (Fig. 7A, top). The two protofilaments A and B form a dimeric structure that assembles into a PHF conformation (Fig. 7A, bottom). An 820 ns classical MD simulation of one single tau protofilament in water showed that the overall fold of the protofilament remained relatively stable during the MD simulation (Figs. S6–S10). Interestingly, the analyses revealed the presence of three cryptic pockets with channel-like features (Fig. 7B and Table S6; pocket 1 (P1) formed by residues Asp314, Glu372, His374, Lys370, Pro312 and Val313; P2 by residues Asn359, Asp358, Gly333-335, Leu357, Pro332, and P3 by residues Ser320, Lys321, Cys322, Gly323, Val363, Pro364, Gly365, and Gly366). These pockets were not present in the initial cryo-EM structure 5O3L[49] and to the best of our knowledge have not been described previously.

The identified pockets present channel-like features with a non-regular shape, as their cavities cross the entire fibril axis longitudinally. This allows water and possibly ions to permeate through the protofilament (Fig. 7C). P2 has the highest water permeability with an average number of water molecules of 9 ± 2, followed by P1 (7 ± 2) and P3 (4 ± 3). P1 showed significantly larger volume and SASA values compared to P2 and P3 in the MD simulation of protofilament A. The volumes of the identified cryptic cavities were 479 ± 156 $Å^3$, 314 ± 106 $Å^3$, and 304 ± 130 $Å^3$ for P1, P2 and P3, respectively (see Table S6 and Figs. S11–S13 in the Supporting Information). Moreover, P1 showed lower volume and SASA variability in the MD simulation of protofilament A, compared to the corresponding pockets in PHF (see panel c) in Figs. S11–S12 and panel d) in Fig. S13). This was likely due to the generally high structural fluctuations observed in these sequences, in particular the abundance of conformationally flexible glycine residues (see Figure B and Fig. S14 in the Supporting Information).

Pocket 2 (P2) was the closest cavity to the dimerization interface of the PHF dimerization region (Fig. 7A, B). Furthermore, this pocket showed the highest druggability as predicted by fpocket[51] (Fig. S15). P2 thus seems particularly attractive for an interaction with PHOX15. Geometrically, the radius of pocket 2 monitored along the MD simulation was 2.1 ± 0.6 Å (Table S7 and Fig. S16), which compares well with that of PHOX15 of 2.4 ± 0.4 Å. This finding, together with the linearity of both PHOX15 and pocket 2, suggest possible steric complementarity. Conversely, the PHOX compounds with bulkier benzyl-benzene moieties would not be sterically able to enter this channel-like pocket.

We repeated the same analyses on a 620 ns MD simulation of the dimeric PHF structure of 5O3L (Figs. 7A and S17), obtaining similar results. Again, we identified P1, P2, and P3 on one protofilament

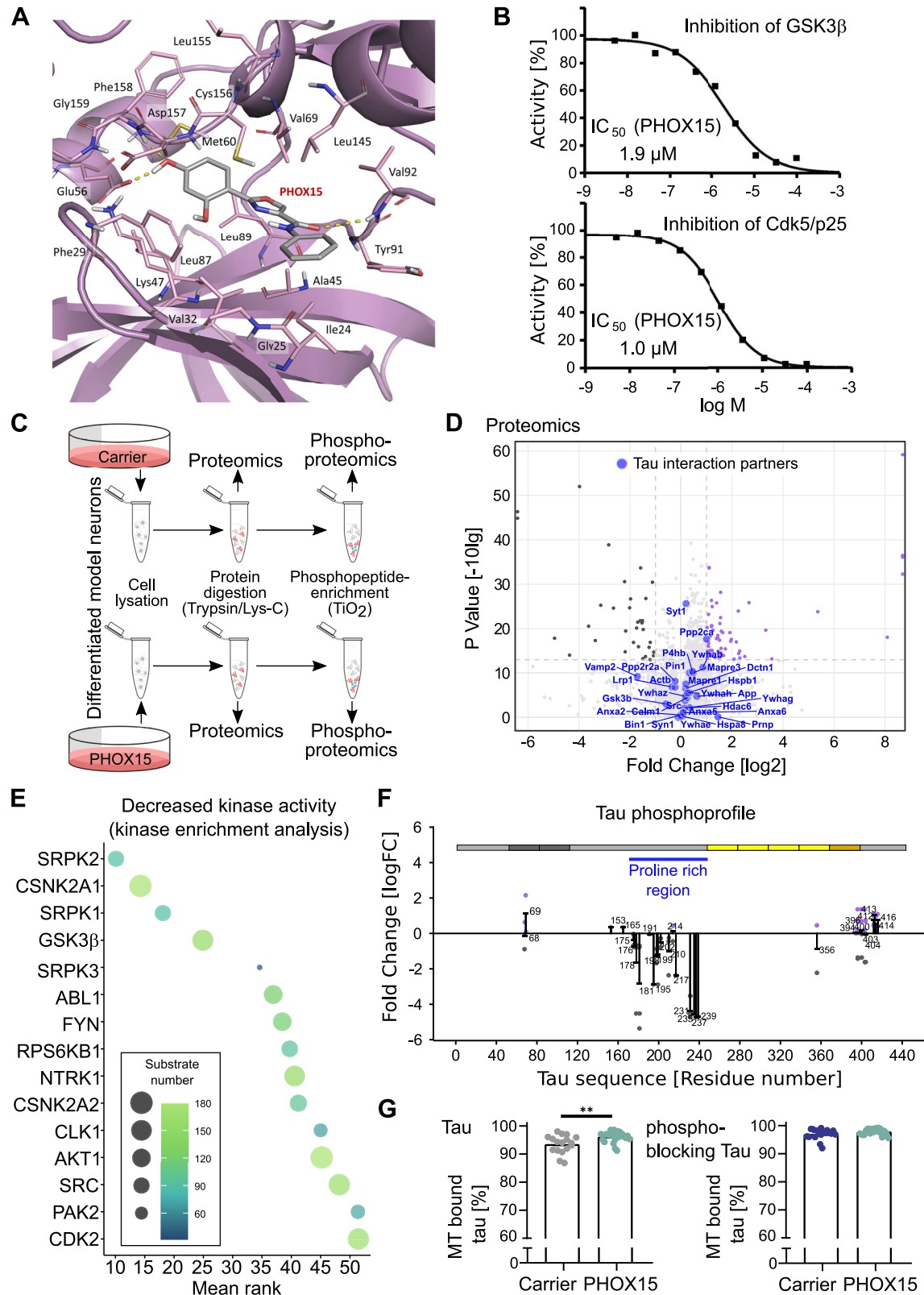

(protofilament A, Table S7 and Fig. S17) and the corresponding pockets P1*, P2*, and P3* on protofilament B. An additional pocket, P4*, was detected in protofilament B, while it was not present in protofilament A. This additional pocket may be due to the observed higher structural flexibility of protofilament B compared to protofilament A (Fig. S6). The radius of P2/P2* on the PHF dimer was similar to that of the single protofilament, with P2 and P2* sharing the same cylinder-like and

smooth cavity with an average radius of $2.2 \pm 0.4$ Å and $2.7 \pm 0.7$ Å, respectively (Table S7 and Fig. S16). Again, the slight difference between P2 and P2* is likely due to the higher flexibility of protomer B (Fig. S6).

As mentioned, MD simulations were also performed starting from a 4R tau isoform structure (PDB ID: 7P65), which includes residues from 272 to 381[11], highlighting remarkable structural differences with

**Fig. 6 | PHOX15 inhibits the tau kinases GSK3β and Cdk5, changes the kinome activity of the cells, and reduces tau phosphorylation at disease-relevant sites.** **A** Binding mode of PHOX15 in the ATP pocket of GSK3β based on docking analysis using available crystal structures. **B** In vitro kinase assays for inhibition of GSK3β and Cdk5 activity by PHOX15 indicate $IC_{50}$ values in the low micromolar range. **C** Schematic representation of the approach for proteomics and phosphoproteomics analyses of differentiated model neurons treated for 20 hrs with PHOX15 (25 μM) or carrier (0.125% DMSO). **D** Volcano plot showing up- or down-regulated proteins in PHOX15-treated cultures compared to controls. None of the >60 experimentally confirmed tau interaction partners[38] were significantly up-or down-regulated. **E** Kinase enrichment analysis of the phosphoproteomic data to identify the pattern of kinases responsible for increased phosphorylation of the cellular

proteins as response to PHOX15 treatment. **F** Phosphoprofile of endogenous tau as determined by phosphoproteomics. The changes in the phosphorylation of individual sites as a response to treatment with PHOX15 are displayed and indicate strong decrease of phosphorylation at several sites in tau's proline rich region. **G** Scatterplots showing percentage of wild-type tau and phosphoblocking tau bound to microtubules as determined by FDAP experiments with the respective PAGFP-tagged tau constructs. Cells were treated with PHOX15 (25 μM) or carrier (0.125% DMSO) for 20 hrs prior to imaging. Mean ± SEM of 26, 22 (PHOX15) and 17, 18 (carrier) for wild-type tau and phosphoblocking tau expressing cells, respectively, are shown. Statistically significant differences as determined by unpaired two-tailed Student's t-tests with Welch correction are indicated. **p < 0.01 (p = 0.005). Source data are provided as a Source Data file.

respect to the PHF conformation described above. In this case, we observed a single channel-like cavity (named P5 in Fig. 7D) that still lies close to pocket P2 of the PHF fold (see Fig. 7B and below), but having nearby residues arranged in significantly different ways due to the marked difference between the two folds. An 800 ns MD simulation of 4R tau (Fig. S18) performed with identical settings did not reveal additional pockets, except for the P5 channel highlighted in Fig. 7D. Interestingly, both P2 of 4R-3R tau and P5 of 4R tau present shape and binding features potentially able to bind PHOX15. Therefore, they both constitute suitable candidates for further evaluation.

### The simulations indicate that PHOX15 disrupts the propensity of glycine triads to form a PHF-like assembly

As described above, P2 of the PHF fold is located at the PHF dimerization interface (Fig. 7A, B). This pocket contains a cluster of three glycine residues belonging to the R3 segment (Figs. 7A and S19A), which we termed the glycine triad. According to our MD simulations of dimeric tau, these glycines and the nearby residues are likely to play a key role in assembling and stabilizing the PHF interface, mainly through hydrogen bonding and salt bridges (Fig. S19). In the identified protofilament A–protofilament B contacts (Table S8), a double H-bonding interaction was established through the backbone carbonyl and amide nitrogens of the central glycine residues (Fig. S19). In addition, salt bridges between the charged side chains of Lys331 and Glu338 residues (5O3L numbering) and hydrogen bonding between the side chain of Gln336 of both protofilaments with the backbone carbonyl residue of Lys331 or Pro332 residues of the other protofilament were observed. These interactions were present in both tau filaments. Protofilament A – protofilament B contacts mediated by the glycine triads have occupancies in the order of 50-85%, while those mediated by the second group of residues had occupancies of 10-67% during the simulation (Table S8), which highlights the key role of the central glycine residues of the two protofilaments in assembling the PHF structure. The upstream (Gly333) and downstream (Gly335) glycine residues (Fig. 7A) form a network of H-bonds with the backbones of Pro332 and Gln336, respectively, placed on the top strand of the protofilament (Table S9). These interactions have occupancies of 80% and appear to be important to stabilize the conformation of Gly334 in a PHF-like arrangement, as highlighted by the dramatic decrease in glycine-proline and glycine-glutamine intra-protofilament H-bonds in the simulation of protofilament A (Table S10).

We set out to assess and compare the Ramachandran plots (Φ and ψ angles) of all Gly-Gly-Gly residues along the MD simulation of dimeric and monomeric (single protofilament) PHF tau, both with and without bound PHOX15. As expected, the conformational freedom of the glycine triads in the simulation of monomeric 4R-3R tau (PDB ID: 5O3L) was significantly higher with respect to that of dimeric 4R-3R tau, the latter exploring only one or two states (Fig. S14). These results are consistent with the tight H-bonding network observed in the simulation of the PHF dimer (Fig. S19). Details about the Φ and ψ angles and hydrogen bonding calculations are reported in the Supporting Information. As expected, the percentages of occurrence of

glycine residues in the PHF-like conformation were very high (around 100%) in the MD simulation of the PHF dimer (see the percentages of occurrence of tau conformations with Φ and ψ angles typical of the PHF-like conformation reported in Table 1).

In contrast, these percentages decreased to 50-60% in the simulation of protofilament A of monomeric tau, indicating that about half of the conformational space of the single protofilament is not prone to dimerization. As for monomeric 4 R tau (PDB ID 7P65), the percentages of glycine triads with Φ and ψ angles in PHF-like conformation and corresponding H-bond interactions were even lower than those observed in both monomeric and dimeric 4R-3R tau (Tables 1 and S11). This finding agrees well with the fact that the 4 R isoform is representative of an earlier conformational status of aggregated tau.

To assess the effects of PHOX15, we then performed docking calculations using the IFD protocol (Schrödinger suite 2022-1)[52] into the characterized P2 and P5 pockets (Fig. 7B, D). PHOX15 could accommodate itself longitudinally along both channels, resulting in favorable hydrogen bonding and hydrophobic interactions (Fig. S20A and B). A series of MD simulations starting from these tau/PHOX15 complexes were then performed to evaluate whether the compound could influence the propensity of the glycines triads to adopt a PHF-like conformation. Interestingly, monomeric 4R-3R tau in complex with PHOX15 showed a further decrease of the percentages of occurrence of PHF-like conformations, which were significantly lower than those observed in the simulations of both monomeric and dimeric tau in the apo form (Table 1 and Fig. S21). Such an effect was even more evident when the analysis included the % of occurrence of tau conformations with the hydrogen bond network typical of the PHF fold (Fig. S19) (Table 1).

Remarkably, the binding of PHOX15 to the P5 pocket of 4 R tau reduced even further the ability of the glycine triads to adopt a PHF-like conformation, the corresponding percentages being around 0% for most filaments (Table 1). The distributions of Φ and ψ angles of the glycine triads resulting from the MD simulations of 4R-3R and 4 R tau in complex with PHOX15 (Fig. 7E) were also substantially different. In particular, while the distribution of Φ and ψ angles was significantly narrow in the MD simulation of dimeric PHF tau, which is indicative of low fluctuations, a higher variability and significantly different values of Φ and ψ angles of the glycine triads were observed in the MD simulations of 4R-3R and 4 R tau in complex with PHOX15 (Fig. 7E). These considerations stand also for the H-bond network of interactions in correspondence of the glycine triads, which is rarely in a PHF fold-like configuration when PHOX15 binds to the P5 pocket of 4 R (Table 1).

Taken together, these results suggest that binding of PHOX15 is a highly dynamic process that affects the ability of the glycine triads to enable tau to progress into a dimerization-prone, advanced stage aggregation state. Such effects are even more pronounced in the earlier stages of tau oligomerization, in agreement with the experimental data presented in this work. As a final note, docking and MD simulations of PHOX15 complexes with tau dimers in PHF conformation showed that the compound was no longer able to influence its

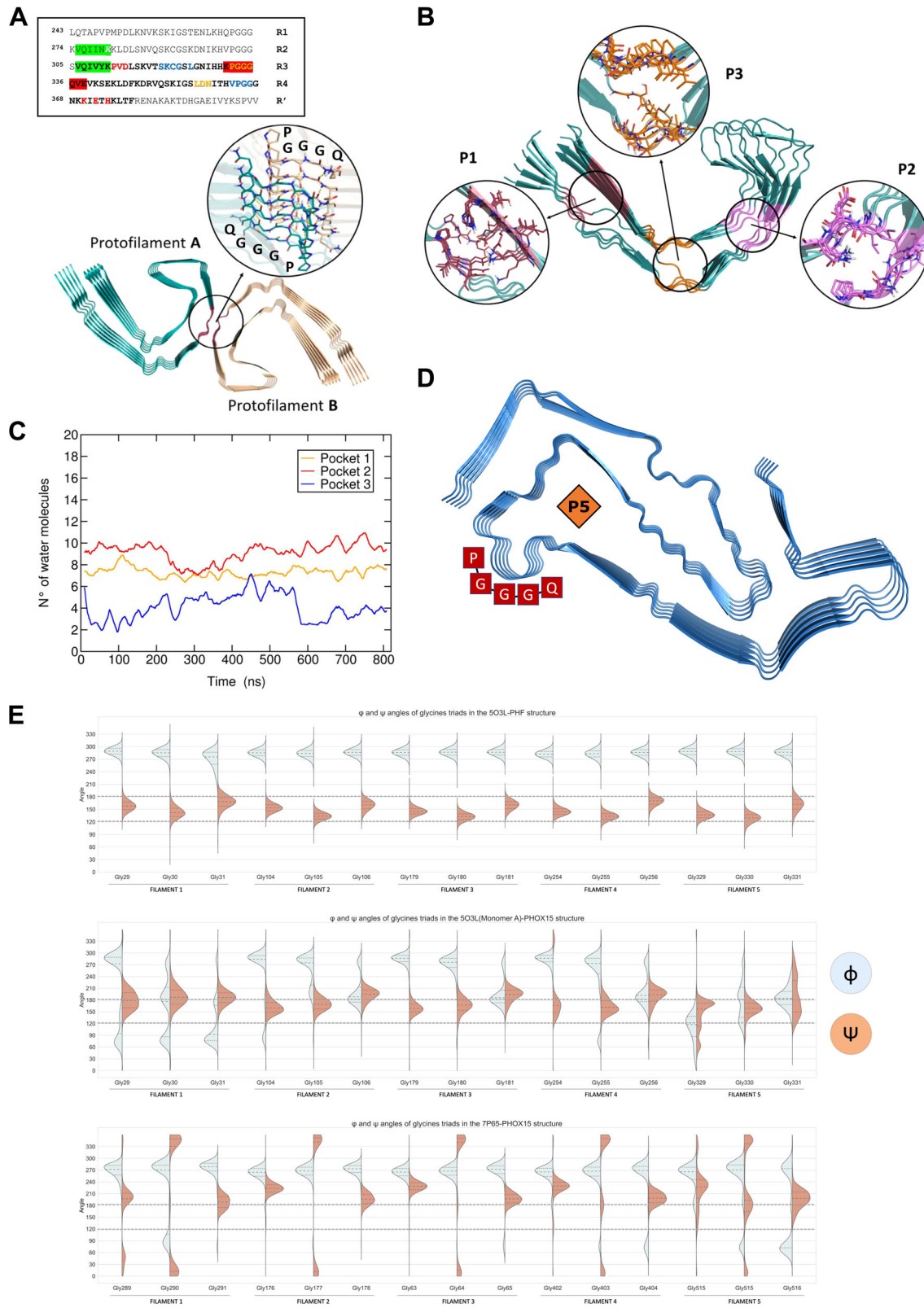

conformation, suggesting that once the PHF dimer is formed, the dimer is too stable to be affected by the compound under investigation.

## Discussion

Tau aggregation and impaired microtubule interaction are believed to play a central role in the development of AD and other tauopathies.

Therefore, drugs that inhibit tau aggregation or restore the interaction of tau with microtubules could be a promising approach to combat tau-induced neurodegeneration. In fact, several drug candidates aiming to inhibit tau aggregation, reduce tau phosphorylation, or decrease tau expression are under investigation[8]. However, mechanism-based drug assays are difficult to establish due to the lack of cell-based models that allow to monitor tau aggregation and effects on tau-

**Fig. 7 | Molecular dynamics simulations highlight cryptic channel-like pockets crossing tau protofilaments. A** Cryo-EM structure of the PHF dimer. Glycine triads are highlighted in red. A close-up view of the glycines interacting between the protofilaments of the PHF is also shown. The sequence of the MT-binding repeat region of the longest Tau isoform containing four repeats (R1, R2, R3 and R4) is shown above. Residues elucidated by cryo-EM (PDB ID: 5O3L)[49], which form the PHF core are shown in bold. Aggregation-prone sequences PHF6 and PHF6* are highlighted in green and the key lysine residue of the aggregation-prone ΔK280 mutant is shown in white italic character. Residues that form the protofilament-protofilament dimerization site are highlighted in red. Residues forming pockets P1, P2 and P3 are shown in red, yellow and blue, respectively. **B** Snapshot of protofilament A after 200 ns of MD simulation. Pocket P1, P2 and P3 are highlighted. **C** Time evolution of the number of water molecules permeating pockets P1, P2 and P3. Standard deviations are not shown for clarity. **D** Structure of 4R tau from PSP. The residues involved in the formation of the PHF core are highlighted with red boxes. The central channel-like P5 pocket is highlighted with an orange diamond. **E** Molecular dynamics simulations of 4R-3R and 4R tau in complex with PHOX15 highlight different distributions of the Φ and ψ angles of the glycine triads with respect to a PHF-prone dimeric conformation. Violin plots showing the distribution of the Φ (light blue) and ψ (light orange) angles of glycine triads in the 4R-3R (PDB ID: 5O3L) dimeric structure (top panel), in the monomeric protofilament A (PDB ID: 5O3L) with PHOX15 bound (middle panel), and protofilament of 4R tau (PDB ID: 7P65) with PHOX15 bound (bottom panel) observed from the MD simulations. Φ and ψ angles of glycine triads are scaled from 0 to 360 degree to facilitate visualization and comparison.

## Table 1 | Percentages of occurrence of the glycine triads of each tau filament adopting a PHF-like conformation

| Tau monomers | 5O3L | | | 7P65 | |
|---|---|---|---|---|---|
| | PHF (%)* | Prot. A (%)* | Prot. A + PHOX15 (%)* | Prot. (%)* | Prot. + PHOX15 (%)* |
| Filament 1 | 86.9 (49.2) | 12.4 (10.1) | 1.2 (0.6) | 5.7 (0.3) | 0.6 (0.1) |
| Filament 2 | 97.0 (55.2) | 57.5 (38.4) | 29.6 (12.4) | 8.4 (0.5) | 0.01 (0.0) |
| Filament 3 | 99.1 (55.5) | 63.5 (44.5) | 23.7 (9.6) | 4.1 (0.8) | 0.1 (0.01) |
| Filament 4 | 98.1 (55.8) | 53.3 (37.4) | 37.4 (17.7) | 18.6 (0.2) | 7.9 (2.3) |
| Filament 5 | 96.8 (55.4) | 11.7 (5.86) | 0.6 (0.01) | 11.5 (2.1) | 11.2 (3.7) |

* The percentages are calculated according to the formula: (number of MD frames with Φ and ψ angles of the three glycine residues in the PHF conformation/total number of frames of the MD simulation) ∘ 100. Numbers in brackets report the percentage of the glycine triads of each tau filament adopting a PHF-like conformation, as well as performing the H-bond interactions of the PHF fold as described in Fig. S19.

The results are reported for the MD simulations of the 5O3L PHF dimer, 5O3L protofilament A alone, the 5O3L protofilament A with bound PHOX15, and the 7P65 protofilament alone and the 7P65 protofilament with bound PHOX15. Numbers in the table refer to the percentages of occurrence of conformations in which the three glycine residues have Φ and ψ angles typical of the PHF fold.

microtubule interaction of the full-length tau protein. While previously "biosensor cells" were generated to probe pathological tau seeds, these cells are based on the expression of only a fragment of the tau repeat domain that does not contain additional regulatory regions such as the proline-rich region[53,54]. The presence of additional regions is of particular importance for the analysis of the role of tau in the disease process, because the tau region involved in mediating tau aggregation and its binding to microtubules overlaps, and regions flanking the microtubule-binding repeats of tau can modulate both activities[49,55,56]. Therefore, approaches aiming to inhibit tau aggregation run the risk of also negatively affecting the physiological interaction of tau with microtubules.

In addition, increasing evidence suggests that soluble oligomeric tau species that precede the formation of neurofibrillary tangles (NFTs), are the toxic species[57–59]. Indeed, NFTs may actually be neuroprotective by sequestering tau[60], which would require the development of tau aggregation inhibitors that selectively reduce the amount of oligomeric tau species without affecting the existing NFTs. Experimental evidence also suggests that disease-like hyperphosphorylation of tau and microtubule binding play a crucial role in the regulation of tau toxicity[13,61], implying that it is important to differentiate the effect of potential drugs on tau aggregation, tau phosphorylation and tau-microtubule interaction in order to develop approaches that positively modulate certain features of tau toxicity.

The multiple facets of tau's involvement in the neurodegenerative cascade also make tauopathies an attractive target for a poly-pharmacological approach, which has been recently pursued in the search of innovative therapeutics for the treatment of such diseases[62]. This approach has gained even more attention with the recent rise in use of computational methods, which can offer the ability to predict the activity profile of ligands in the iterative design and optimization steps of a preclinical drug discovery project[63]. Along these lines, it would be quite attractive to have a drug that simultaneously inhibits tau aggregate formation and reduces tau phosphorylation at disease-

relevant sites, thereby restoring physiological tau-microtubule interaction and regulation of axonal microtubule polymerization.

In this work, (1) we developed a quantitative live-cell imaging approach to analyze tau-microtubule interaction and tau amyloid formation in a cellular tauopathy model; (2) we used the approach to screen a set of candidate molecules that, according to chemoinformatic analysis, are likely to inhibit tau aggregation and disease-associated kinases; (3) we identified one compound, PHOX15, that inhibited tau aggregation in vitro and in cells, restored tau-microtubule interaction of an aggregation-prone tau construct in neurons, and increased tau-microtubule interaction by reducing tau phosphorylation at disease-relevant sites; and (4) we found cryptic pockets in tau protofilaments with channel-like features through molecular dynamic simulations where PHOX15 might bind and decrease the ability to adopt a PHF-like conformation.

Our data indicate that PHOX15 normalizes the tau-microtubule interaction through two distinct mechanisms (Fig. 8). On the one hand, it increases the interaction of an aggregation-prone tau construct with microtubules to a physiological level by reducing the formation of tau amyloid. This activity might be mediated by PHOX15 binding to channel-like pockets crossing tau protofilaments, thereby disrupting tau oligomerization. In this regard, this type of interaction is favorable as it would interfere with tau aggregation but would not compete with tau binding to microtubules. On the other hand, PHOX15 reduces tau phosphorylation at disease-relevant sites, particularly in the proline-rich region, which contains a cluster of phosphorylation sites that can be phosphorylated by GSK3β. Indeed, most sites in this region that show more than two-fold reduced phosphorylation are predicted phosphorylation sites for GSK3β (i.e., Ser235, Ser237, Thr231, Ser195, Thr181, Thr217, Ser198, Ser199, Ser210; NetPhos-3.1b prediction). Analysis of kinome activity showed that PHOX15 changes the activity of several kinases in addition to GSK3β, and it has yet to be shown that this does not have any undesirable side effects that would require further drug optimization.

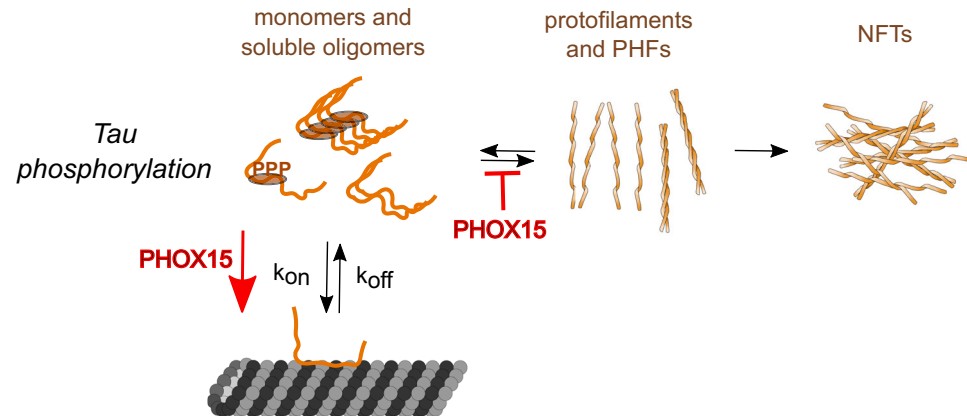

**Fig. 8 | Model showing the effect of PHOX15 on tau aggregation.** Our data indicate that PHOX15 reduces the formation of tau filaments and decreases the phosphorylation of tau at disease-relevant sites. Both activities increase the interaction of tau with microtubules to a physiological level.

Our PC12 cell data show that the aggregation-prone tau construct does not form insoluble tau aggregates in the cells and cells transiently expressing the TauΔK280 construct were negative for staining with Amytracker™, an optotracer that binds to amyloid fibrils containing at least eight repetitive beta-sheets (Ebba Biotech, Solna, Sweden). Lentiviral infection of DRG neurons allowed determination of the long-term effects of tau expression in authentic neurons. In these experiments, we observed an increased formation of Amytracker™-positive tau amyloids in neuronal cell bodies over time, which became significant after two weeks of continuous TauΔK280 expression. Increased tau amyloid formation in the cell body correlated with decreased levels of tau binding to microtubules in the axon, which indicates progressive formation of soluble tau oligomers that impair tau-microtubule interaction. The data also indicate that PHOX15 reduces the formation of soluble tau oligomers and effectively increases the availability of tau to interact with microtubules.

Although PHOX15 needs further testing in a systemic setting, many previous drug studies were initiated without a thorough mechanism-based analysis of their functions, which is particularly critical for a complex target like tau with many interaction partners and regulation by various post-translational modifications[9,64]. We are confident that our cell model to study the effect of drugs on the modulation of tau aggregation and tau-microtubule interaction and the identification of the small molecule PHOX15 as a promising polypharmacological drug candidate will help advance approaches that specifically target the key traits of tauopathies. Furthermore, we are confident that the simulations provide the basis for a structure-based optimization of the activity of these compounds as well as the rational design of new anti-aggregation tau ligands.

## Methods
### Expression vectors and virus preparations
Prokaryotic expression plasmids were based on human adult tau (Tau441wt) in a pET-3d vector[65]. Eukaryotic expression plasmids for tau variants were based on human adult tau (Tau441wt) with an amino-terminally fused PAGFP tag in a pRc/cytomegalovirus (CMV) expression vector[66]. The ΔK280 deletion was introduced into the expression plasmids by site-directed mutagenesis using phosphorylated primers (5′AAGCTGGATCTTAGCAACGTC and 5′ATTAATTATCTGCACCTTCCCG CC) with Platinum SuperFi DNA Polymerase (ThermoFisher Scientific, USA). Only the phosphoblocking tau mutant and the corresponding control for Fig. 6G were based on the 352 amino acid fetal isoform of tau and was created by changing the codons for Ser198, Ser199, Ser202, Thr231, Ser235, Ser396, Ser404, Ser409, Ser413, and Ser422 in GCT (alanine) as previously described[67]. Lentiviral vectors for Tau441wt and TauΔK280 with amino-terminally fused PAGFP tag were constructed in

L22FCK(1.3)GW (provided by P. Osten, Northwestern University, Chicago, IL) containing the neuron-specific promoter α-CaMKII[68]. Sequences introduced by PCR were verified by DNA sequencing (Seqlab-Microsynth, Göttingen, Germany). For the production of lentivirus, 293FT human embryonic kidney cells (Thermo Fisher Scientific, USA) were transfected with the expression vector and two helper plasmids and viral particles from the supernatant were concentrated by ultracentrifugation as previously described[69].

### Cell lines
PC12 cells (originally obtained from J.A. Wagner, Harvard Medical School) were cultured in serum-DMEM (DMEM supplemented with 10% fetal bovine serum and antibiotics (100 U/ml penicillin and 100 μg/ml streptomycin)), at 37 °C with 10% $CO_2$ in a humidified incubator and transfected with Lipofectamine 2000 (Thermo-Fisher Scientific, USA) as previously described[13]. After transfection, the medium was replaced with serum-reduced DMEM (DMEM supplemented with 1% fetal bovine serum and antibiotics) and the cells were neuronally differentiated for 4 days by the addition of 100 ng/ml 7 S mouse NGF (Alomone Labs, Israel).

### Primary neuronal cultures
C57BL/6 J mice were kept and killed in accordance with the German animal care regulations based on the FELASA guidelines. The housing rooms were kept at a 12:12 light-dark cycle with a constant room temperature of $22 \pm 2$ °C and relative humidity of $55 \pm 10$%. Three-month- or one-year-old mice were sacrificed by cervical dislocation. The spine was quickly dissected and the spinal cord exposed. Dorsal root ganglia (DRGs) were extracted, dissociated and enzymatically digested essentially as previously described[70]. The resulting suspension, containing isolated DRG neurons, was plated in DRG culture medium (Neurobasal A supplemented with 2% B-27, 1% fetal bovine serum, 1% horse serum, 20 μM L-glutamine, 0.1% βME, 100 μg/ml Primocin®) supplemented with 100 ng/mL 7 S mouse NGF. Infectious lentiviral particles were applied on day 3. After 6 hours of incubation, the medium was replaced with fresh medium containing NGF, and incubation was continued at 37 °C with 5% $CO_2$ in a humidified incubator. Live cell imaging was performed as indicated.

### Chemicals and cell treatments
The 2-phenyloxazole (PHOX) derivatives[28] were prepared as 20 mM stock solutions in DMSO and were stored at −20 °C in light-protected Eppendorf cups.

### Measurements of neuronal viability
Metabolic activity and viability were assessed using a combined MTT/LDH assay. PC12 cells were cultured in 96-well plates at 10,000 cells per

well in 50 µl serum-reduced DMEM supplemented with 100 ng/ml 7 S mouse NGF. After 48 hours, test compounds at final concentrations of 0, 6.25, 12.5, 25, 50 and 100 µM were added in a further volume of 50 µl and incubation was continued for 20 hours. For LDH measurements, 50 µl from each well was transferred to a new 96-well plate and 50 µl LDH-reagent (4 mM iodonitrotetrazolium chloride (INT), 6.4 mM beta-nicotinamide adenine dinucleotide sodium salt (NAD), 320 mM lithium lactate, 150 mM 1-methoxyphenazine methosulfate (MPMS) in 0.2 M Tris-HCl buffer, pH 8.2) was added. The plate was shaken for 10 seconds and incubated in the dark for 10 minutes. Absorbance was measured at 490 nm using a ThermoMax Microplate Reader operated with SoftMaxPro Version 1.1 (Molecular Devices Corp., Sunnyvale, USA.). Measurements were normalized to a positive control to which 1% Triton X-100 had been previously added. For MTT measurements, 50 µl of MTT-reagent (2 mg/ml MTT (3,(4,5.dimethylthiazol-2-yl)2,5-diphenyltetrazolium bromide in prewarmed serum-reduced DMEM) was added to each of the remaining wells. Cells were incubated for 2 hours at 37 °C and the reaction was stopped by the addition of 50 µl lysis buffer (20% (wt/vol) sodium dodecyl sulfate in 1:1 (vol/vol) N,N-dimethylformamide/water, pH 4.7). After overnight incubation at 37 °C, optical densities at 570 nm were determined. Measurements for MTT conversion were normalized to optical densities of the negative control wells. All experiments were performed in triplicates in two independent plates.

### Purification of recombinant tau protein
pET-3d-tau plasmids were transformed into Escherichia coli BL21(DE3) pLysS cells for expression, and the cells were grown, induced and harvested as previously described[65]. Tau was purified from the cell extract of 200 ml of culture by sequential anion exchange and phosphocellulose chromatography as previously described[65]. Tau protein eluate was dialyzed against PBS containing 2 mM $MgCl_2$, concentrated with Vivaspin® (15 R, 2,000 MWCO, Sartorius, UK) and adjusted to 1 mM DTT. Protein concentrations were determined by densitometry of Coomassie Brilliant Blue-stained gels using bovine serum albumin as a standard.

### Tau aggregation assay
Assays were performed with 0.5 mg/ml recombinant tau protein in 30 mM MOPS/NaOH (pH 7.4) containing 200 µg/ml heparin and 1 mM 4-(2-aminoethyl)benzenesulfonyl fluoride (AEBSF, Applichem, Darmstadt, Germany) in an assay volume of 25 µl essentially as previously described[67]. Incubation was at 37 °C for the indicated times and concentrations. In order to quantify the amount of aggregates, the mixture after the assembly reaction was centrifuged for 1 h at 100,000×g and 4 °C. One third of the supernatant and pellet fractions were separated by SDS-PAGE with 15% acrylamide, stained with Coomassie Brilliant Blue and quantitated by densitometry.

### Electron microscopy
For analysis by transmission electron microscopy (EM), aggregation assays were performed with 0.4 mg/ml recombinant TauΔK280 in an assay volume of 25 µl for 24 hrs at 37 °C. To test for drug induced disassembly, PHOX15 or carrier were added in 10% of the volume, mixed, and incubation continued for another 24 hrs. The samples were sonicated for 2 min twice with a pause of 1 min. Pioloform-coated and glow-discharged grids were floated on a drop of the sample for 10 min, washed with water, negatively stained with 1% uranyl acetate, and dried. EM was performed with a Zeiss 10CR electron microscope at 60 kV.

### Immunoblotting
After separation by SDS−PAGE, proteins were transferred to Immobilon-P PVDF membranes (Millipore, USA), followed by immunoblotting. Detection employed PHF1 antibody (mouse monoclonal

IgG1[71]; kind gift of Peter Davies, Albert Einstein College of Medicine (New York, USA)) and peroxidase-conjugated AffiniPure donkey anti-mouse IgG (H + L) secondary antibody (Jackson ImmunoResearch Laboratories, Inc., USA; code number: 715-035-151; dilution 1:10,000). Protein bands were detected using enhanced chemiluminescence with SuperSignal West Dura extended duration substrate (Thermo Fisher Scientific, USA) according to the manufacturer's protocol. Quantification of the blots was carried out with Gel-Pro Analyzer 4.0 (Media Cybernetics L.P., USA) or with FusionCapt Advance (Vilber Lourmat, France).

### Live-cell imaging and Fluorescence Decay After Photoactivation (FDAP)
For FDAP experiments, wild-type tau or TauΔK280 expressing cells were plated on 35-mm poly-L-lysine and collagen-coated (for PC12 cells) or laminin-coated (for DRG neurons) glass-bottom culture dishes (MatTek, USA). PC12 cells were neuronally differentiated after transfection by medium exchange for serum-reduced DMEM containing 100 ng/mL 7 S mouse NGF. Cultivation was continued for 4 days with medium exchange for serum-reduced DMEM containing NGF and without phenol red one day prior to live imaging. Treatment of the cells was performed 20 hours prior to live imaging by addition of the respective compound (or DMSO for carrier control) in the desired concentration. Live cell imaging for photoactivation experiments was essentially performed as described previously[15] using a laser scanning microscope (Nikon Eclipse Ti2-E (Nikon, Japan)) equipped with a LU-N4 laser unit with 488-nm and 405-nm lasers and a Fluor 60× ultraviolet-corrected objective lens (NA 1.4) enclosed in an incubation chamber maintaining 37 °C and 5% $CO_2$. Photoactivation of a 6 µm long neurite segment was performed with a 405-nm laser using the microscope software (NIS-Elements version AR 5.02.03 (Nikon, Japan)). A set of consecutive image series (time stack) was obtained at a frequency of 1 frame/s, and 112 frames were collected per activated cell at a resolution of 256 × 256 pixels. Effective diffusion constants and fraction of microtubule-bound tau were determined by fitting the fluorescence decay data from photoactivation experiments using a one-dimensional diffusion model function for FDAP, as previously described[25]. For staining with optotracer, Amytracker 680 was diluted 1:500 in the culture medium. After 30 minutes of incubation, cells were imaged with a 453-nm laser. Fluorescence intensity was quantified using Fiji[72] by determining corrected total cell fluorescence (CTCF) according to the formula: CTCF = integrated density − (area of selected cell × mean fluorescence of background readings).

### Proteome and phosphoproteome analysis
PC12 cells were neuronally differentiated by culture for 4 days in serum-reduced DMEM containing 100 ng/ml 7 S mouse NGF. 20 hours prior to sample preparation, cells were treated with 25 µM PHOX15 or carrier (0.125% DMSO). Cells were washed in ice-cold PBS, incubated in lysis buffer (8 M urea in 50 mM Tris/HCl, pH 7.8) supplemented with Phos-Stop tablets (Roche Diagnostics GmbH, Germany) and sonicated for 5 cycles of 5 s each at 10% amplitude (Branson Digital Microtip Sonifier 250-D, Connecticut, USA). Samples were cleared by centrifugation at 4 °C and 23,000×g for 30 min and protein concentration determined using Pierce™ BCA Protein Assay (Thermo Fisher Scientific, USA). 0.2 µg/µl α-casein were added to a protein amount of 1.2 mg, and the reduction and alkylation were carried out in lysis buffer containing 15 mM iodoacetamide and 5 mM DL-dithiothreitol. Proteins were digested for 18 h with trypsin/Lys-C Mix (Promega Corporation, USA) and 10 µg of each sample was used for proteome analysis. For phosphoenrichment, remaining samples were desalted using Sep-Pak® Classic C18 cartridges (Waters, Ireland) that had been prewashed and equilibrated with 5 ml methanol, 5 ml 80% acetonitrile and 2×5 ml 0.5 % formic acid. After sample application, the Sep-Pak was washed once with 5 ml of 0.5 % formic acid and eluted with 4 ml of 80% acetonitrile/

0.5 % formic acid. The eluate was lyophilized and enriched with a High-Select™ TiO2 Phosphopeptide Enrichment Kit (Thermo Fisher Scientific, USA). For proteome and phosphoproteome analysis, samples were taken from a PepMap C18 easy spray column (Thermo Fisher Scientific, USA) with a linear gradient of acetonitrile from 10–35% in $H_2O_2$ with 0.1% formic acid for 180 min at a constant flow rate of 25 nl/min. MS analysis was performed as previously described[73]. The *.raw data files were analyzed with PEAKS Online software (Bioinformatic Solutions Inc, Canada). PEAKS Q (de-novo-assisted quantification) analysis was used for data refinement with mass correction, de-novo sequencing and de-novo-assisted database search, and subsequent label-free quantification. The search engine was applied to Rattus norvegicus *.fasta databases. MS/MS searches were performed using a mass tolerance of 10 ppm parent ions and a mass tolerance of 0.2 Da fragments. Trypsin with up to two missing cleavage points was selected as the cleavage enzyme. Carbamidomethylation modification was chosen as the fixed modification and the oxidation of methionine, acetylation of lysine and phosphorylation of serine, threonine and tyrosine were chosen as the variable modifications. A maximum of three variable modifications were allowed per peptide. Normalization to the total ion current level for each sample was performed. ANOVA test was used to calculate the significance level for each protein, outliers were removed, and the top three peptides were used for protein signal quantification where it was possible. The peptide identification was considered valid at a false detection rate of 1% (q-value < 0.001) (maximum delta Cn of the percolator was 0.05). The minimum length of acceptable identified peptides was set to six amino acids. Each condition was analyzed in triplicate. All proteins were assigned their gene symbol via the Uniprot knowledge database (http://www.uniprot.org/). Tau interaction partners selected for representation on a volcano plot were based on previously described region-specific interactors[9]. Normalized phosphoproteomics data revealed phosphopeptides that were down-regulated after treatment with PHOX15, gene symbols associated with these phosphopeptides were used for kinase enrichment analysis via KEA3 application[42]. Mean rank and the number of substrates that were associated with fifteen highest-ranking kinases were used for representation.

### Statistical analysis

Statistical analysis was carried out with GraphPad Prism v8.0.1 (GraphPad Software, USA). All data sets were tested for normality using D'Agostino-Pearson and Shapiro-Wilk test. If necessary, data sets were log transformed in order to enable further statistic testing. Statistical outliers were identified using the ROUT method. The homogeneity was assessed using Levene's test. An unpaired two-tailed t-test was used to compare two datasets. In the case of unequal variances, the Welch's correction was applied. A one-way ANOVA followed by Dunnett's post-hoc test was performed to compare more than two data sets with a single control. All statistical values are expressed as mean ± SEM.

### Ligand-based and structure-based analyses
See Supporting Information.

### Reporting summary
Further information on research design is available in the Nature Portfolio Reporting Summary linked to this article.

## Data availability
The mass spectrometry proteomics data have been deposited to the ProteomeXchange Consortium via the PRIDE[74] partner repository with the dataset identifier PXD048913 and 10.6019/PXD048913. Scripts used for FDAP analysis as described in[25] are available at https://github.com/Department-of-Neurobiology/. Source data are provided with this paper.

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

## Acknowledgements

We are grateful to Hilmar Bading, Department of Neurobiology and Interdisciplinary Center for Neurosciences, University of Heidelberg, for providing the opportunity to carry out the electron microscopy work in his laboratory. We thank Lisha Singh and Halvard Gremme (Department of Neurobiology, Osnabrück University) for help with DRG neuron culture and tau aggregation experiments and Jürgen J. Heinisch (Department of Genetics, Osnabrück University) for preparation of the TauΔK280 virus DNA. This work was supported by the Deutsche Forschungsgemeinschaft (DFG BR1192/14-1 to RB), Fondo di Ateneo per la Ricerca (FAR 2019, DR496/2019 to GR) and European Union's Horizon 2020 research and innovation program H2020-MSCAITN-2019-EJD - grant agreement no: 860070 (NB, AS, DP and RB).

## Author contributions

Luca Pinzi: Investigation, Writing - Original Draft; Christian Conze: Investigation, Writing - Original Draft; Nicolo Bisi: Investigation, Writing - Review & Editing; Gabriele Dalla Torre: Investigation, Writing - Review & Editing; Ahmed Soliman: Investigation, Writing - Review & Editing; Nanci Abreu: Investigation, Writing - Review & Editing; Nataliya I. Trushina: Investigation, Writing - Review & Editing; Andrea Krusenbaum: Investigation, Writing - Review & Editing; Maryam Khodaei Dolouei: Investigation, Writing - Review & Editing; Andrea Hellwig: Investigation, Writing - Review & Editing; Michael S. Christodoulou: Resources; Daniele Passarella: Resources; Lidia Bakota: Writing - Original Draft, Supervision; Giulio Rastelli: Conceptualization, Writing - Original Draft, Supervision, Funding acquisition; Roland Brandt: Conceptualization, Writing - Original Draft, Supervision, Funding acquisition.

## Funding

## Competing interests

The authors declare no competing interests.

## Additional information

¹Department of Life Sciences, University of Modena and Reggio Emilia, Modena, Italy. ²Department of Neurobiology, School of Biology/Chemistry, Osnabrück University, Osnabrück, Germany. ³Department of Neurobiology, Interdisciplinary Center for Neurosciences, Heidelberg University, Heidelberg, Germany. ⁴Department of Chemistry, University of Milan, Milan, Italy. ⁵Center for Cellular Nanoanalytics, Osnabrück University, Osnabrück, Germany. ⁶Institute of Cognitive Science, Osnabrück University, Osnabrück, Germany. ⁷Present address: Drug Discovery Unit, Wellcome Centre for Anti-Infectives Research, Division of Biological Chemistry and Drug Discovery, College of Life Sciences, University of Dundee, Dundee DD1 5EH, UK. ⁸Present address: Department of Food, Environmental and Nutritional Sciences (DeFENS), University of Milan, Milan, Italy. ⁹These authors contributed equally: Luca Pinzi, Christian Conze. ✉e-mail: giulio.rastelli@unimore.it; roland.brandt@uni-osnabrueck.de

