## [Peer Review File · Nature Communications]

Quantitative live cell imaging of a tauopathy model enables the identification of a polypharmacological drug candidate that restores physiological microtubule interactionReviewers' comments:

Reviewer #1 (Remarks to the Author):

This paper presents potential inhibitors of tau aggregation, 2-phenyloxazole (PHOX) derivatives. PHOX15 is shown to act by two mechanisms; firstly restoring microtubule binding and inhibiting tau aggregation and secondly inhibiting the tau kinases GSK3 β and Cdk5, reducing tau phosphorylation. It uses a nice fluorescence decay after photoactivation assay to measure microtubule binding in cells. Given the role of tau aggregation in neurodegenerative disease the development of new therapies is clearly important. However, I found the logic of the paper hard to follow. There are cell-free experiments using heparin to induce aggregation, cell experiments and then modelling and confusingly these are all appear to be done on different structures of tau filaments meaning that there are some key questions and additional experiments that need to be addressed before publication to support the conclusions about the mechanisms in the paper :

1. Tau exists in 6 isoforms. In AD a mixture of 3R and 4R isoforms aggregate while in tauopathies either 3R or 4R tau aggregates. This paper uses wt 2N4R tau and Tau Δ K280. It also mentions fetal tau (3R tau) in the methods although this does not seem to have been used in the experiments? The structure of heparin induced tau filaments has been shown by Goedert and co-workers to be different from those found in human brain. The filaments are also different between AD and 4R tauopathies where just 4R tau form filaments such as in PSP and CBD. This being the case it is hard to understand why all the modelling as I understand it, is done on AD filaments not 4R filaments from a tauopathy and /or the filaments formed with heparin and the relevance of modelling AD protofilaments to the experimental data ? It seems much more logical to include or focus on 4R protofilaments in this section to model the cell experiments and possibly also include the modelling the heparin induced filaments. Are the same pockets and glycine triads present in 4R filaments and can PHOX15 bind to these?

The modelling of the AD filaments is also potentially interesting if the results with the 4R filaments are similar to AD, since this might show the potential of this approach as a therapy for AD.

2. The aggregation of tau in the presence of heparin is complicated by heparin acting as an inducer and bypassing the nucleation phase of the aggregation reaction. A ThT kinetics assay in the presence and absence of PhOX 15 may provide additional evidence that it is an inhibitor of aggregation and may elucidate its mechanism to complement the data in Figure 3. If this worked a concentration dependent study could be performed.

3. The effect of PHOX on cells is an interesting result showing that this restores microtubule binding. In figure 1F the authors show that the cells expressing Tau Δ K280 develop aggregates over time. It would be very convincing if they could show that the presence of PHOX15 reduces tau aggregation over time in these cells. Is it possible to include this experiment? Without this result it seems that there is a significant gap between the results and developing this work towards a potential therapeutic which significantly lessens the impact of the paper.

Reviewer #2 (Remarks to the Author):

The microtubule-associated protein tau is subject to hyperphosphorylation and forms pathological aggregates that contribute to neurodegeneration. In this study, the authors apply a live-cell imaging approach based on photoactivation to monitor pathological changes of human tau regarding its interaction with microtubules. They identify the small molecule PHOX15 to inhibit kinases that are known to phosphorylate tau. A molecular dynamics simulation further suggests that PHOX15 could bind to critical regions within the protofilament and might inhibit initial steps of tau aggregation.

The study addresses an interesting topic of high relevance to neurodegeneration, however requires major revision and is not suitable for publication in its present form. Many data are based on indirect imaging and do not directly control the binding of tau to microtubules. Similar live cell

imaging assays based on recovery after photobleaching have been previously used in the field. In any case, results from these assays require biochemical control experiments to independently monitor tau-microtubule interactions. The PHOX15 compound is an interesting molecule but seems to have dual effects on kinase activity and protofilament aggregation. Most of the effects are in the range of 10% or less and require further controls.

Specific Points:

1. Title: the term "physiological" regulation is not yet justified. Biochemical experiments to study tau-microtubule interactions are required in neurons.
2. It needs to be shown that tau forms aggregates (e.g. using oligomer-specific antibodies) and that PHOX15 alters these aggregates.
3. The western blot in Figure 1A requires clarification. What is shown in which lane?
4. Figure 1C: The lower right graph is a quantification of microtubule-bound tau. How was it calculated? The live imaging assay measures diffusion but not tau-MT binding. Representative images should be added to the graphs.
5. Figure 1F: Amytracker detects tau aggregates. How did the authors quantify the binding of Tau Δ K280 to microtubules based on immunostaining? Representative images for individual time points and quantification should be included.
6. Figure 2D: the graph quantifies MT bound tau (%). No representative images are shown. Indirect binding/unbinding via diffusion requires an independent binding experiment.
7. Figure 3A, B: is this result restricted to Tau Δ K280? A second tau mutant should be included. Why is wildtype tau detected in aggregates?
8. Figure 3D: Indicate the antibody used for detection. No loading control is shown. Quality of the image is weak.
9. Figure 4: PHOX15 affects GSK3beta, CDK5 (B) and other kinases (E), all of which have multiple substrates. specific is this approach? Are there any side effects following PHOX15 treatment? Control experiments are required to test whether neuronal transmission remains normal
10. Figure 4G: effects are in the range of about 5% and I am surprised that they are highly significant based on the number of n-values. An independent complementary approach is required to confirm the FDAP data. The Y-axis should not be interrupted in graphs.
11. Page 15, line 456 and Table S5: mention whether PHOX15 does cross the blood-brain barrier.
12. Figure 5: This theoretical dataset is very interesting, however no experimental data are included to test this hypothesis. This should be added.
13. PHOX15 seems to have dual effects on kinases and protofilaments. How can this be explained? Which effect is dominant? Hypo-phosphorylated Tau per se has a lower tendency to form aggregates. Maybe, the proposed loss in Tau aggregation after application on PHOX15 is only/mainly based on the inhibitory effect of PHOX15 on the Tau kinases.
14. PHOX15 should at least be added to neurons (better: injected to brains) derived from an Alzheimer's disease mouse model. Does it affect pathology in cell culture or in vivo?

Reviewer #3 (Remarks to the Author):

The manuscript by Pinzi et al establishes live-cell imaging approach to track colocalization of tau bound with microtubules. By using this model and combining with chemoinformatic analyses and molecular dynamics simulations, the authors report discovery of PHOX15, a small molecule that inhibits tau aggregation, restores the physiological microtubule interaction of tau, and inhibits kinases associated with phosphorylating and inducing tau aggregation. The work is of general interest and importance, however, is lacking in crucial detail and experimental controls which need to be addressed before publication.

1. Binding of PHOX15 to tau is inferred from docking and MD studies and suggests PHOX15 interferes with ability of tau to achieve the PHF conformation. But tau folds in many different ways to achieve fibrils with varied structures. How does binding of PHOX15 affect the variety of possible tau fibril folds? This is especially important since it is likely that the experimental models the authors use could adopt alternate tau fibril polymorphs with structures that differ from the ones seen in AD.
2. In Line 68 the authors say "Mechanisms of toxicity are also controversial, but prefibrillar tau oligomers and soluble tau with disease-like modifications can be toxic". It is surprising there is no

mention here of toxicity by fibrils given that tau inclusions are the observation the authors refer to in Line 60 when justifying tau as a therapeutic target. The Intro would be strengthened by the authors providing some perspective or at least mentioning tau fibrils.

3. How does the delta K280 mutant effect interaction of tau with microtubules, and how can one extrapolate results of this study to AD, which is vastly a disease of non-mutated tau?

4. Loading controls are missing from Fig. 1A.

5. The interpretation of the authors that higher effective diffusion corresponds with decreased microtubule binding is not well supported. The argument could be strengthened by additional control experiments showing what effects microtubule stabilization and disassembly reagents have on FDAP curves in wt and Tau Δ K280 cells. These controls are needed to support any interpretation of the tau localization of microtubules.

6. This reviewer does not follow the logic in Line 127 that increased diffusion indicates reduced tau interaction with microtubules. If tau is nonbound by microtubules, it should have greater diffusion and hence be more difficult to photobleach. Thus, increased decay of Tau Δ K280 would seem to suggest Tau Δ K280 is relatively more so bound to microtubules. And wouldn't FRAP be a more appropriate measure of diffusion and microtubule occupancy since tau that is more fractionally bound to microtubules would recover with slower kinetics?

7. To test the hypothesis that puncta corresponding to aggregated (oligomeric?) tau impedes microtubule binding by tau, the authors measure Amytracker staining 3 weeks after transduction (Line 134). The authors write "aggregates could be observed in most cell bodies". How is the specificity of the Amytracker confirmed? Control experiments showing the time dependent development of the phenotype post-transduction is needed to confirm correlation of Amytracker signal with tau aggregation, as is Amytracker staining in non-transfected cells.

8. Description of the discovery of PHOX15 is lacking in detail. For instance, the authors write "PHOX 5, 8, 15 and 20 were closely related to potent tau aggregation inhibitors". There are no citations given to link the chemical structures shown in Figure S1 with tau inhibition? Please provide additional citations and supporting information.

9. In Figure 2E, what is the right most data point labeled Tau PHOX15? What concentration PHOX15 is it? How does it fit into this figure with comparison to Tau delta K280?

10. The statement in Line 219: "Cylindrical filaments with periodically thin regions are considered to be the PHF precursor" is unclear but seems to draw correlation between recombinant fibrils and PHFs from AD. This is a dangerous correlation to draw given recent studies that show striking differences between recombinant and AD tau fibrils. Without confirmatory data, equating recombinant fibrils and AD PHFs should be avoided.

11. Error bars are missing from Fig. 4b

12. With regard to kinase activity, the authors write: "This binding mode is consistent with the structure-activity relationships (SAR) of other PHOX derivatives tested in this work (data not shown)." What is meant by this? Did experimental screening with other analogs suggest importance for the amide carbonyl, para-hydroxyl, and phenyl rings? If so, these data are crucial validations and should most certainly be shown.

13. Description of molecular docking is absent from the Results section and is inadequately described. The authors write "The pockets were not present in the initial cryo-EM structure 5O3L." Also the authors write "the structure of the protofilament remained relatively stable during the MD." How then were the proposed sites of binding deduced if MD failed to open the proposed sites of binding and the sites are not exposed in the beginning structure used?

14. The authors suggest ligand binding might occur due to channels that open in the tau PHF due to conformational and abundance of conformationally flexible glycine residues. This is not supported by the CryoEM data. Although the PDB file 5o3L used for MD was low resolution, which prohibited measures of conformational flexibility, a later structure 6HRE provides experimental B factors. Residues in P2, namely GLY 333-335 have B factors of 42-46. These are quite low compared with other regions of the PHF where B factors sometimes approach 100. Therefore there is general disagreement of the interpretation of high structural fluctuations at sites with Glycines and pre-existing experimental data.

15. It seems unlikely that any PHOX analogs enter cavities in the PHF. This is even alluded to by the authors who write that there is no effect seen with incubation of PHOX with PHFs or recombinant fibrils. A more likely mechanism that is proposed by the authors is PHOX binds tau monomers preventing PHF folds from emerging since interaction of PHOX in the cavities would prevent folding of tau into the PHF conformation.

Minor Issues

16. The phrasing: "quantitatively monitor pathological changes of human tau as it interacts with axonal microtubules" in the Abstract reads in a confusing way. Human tau as it interacts with axonal microtubules is a physiological interaction. Dissociation from microtubules might be considered pathologic. Please try to revise the sentence to more clearly articulate these distinctions for the reader.

17. In Line 73, it is said "Because abnormal forms of tau could trigger a plethora of pathomechanisms". Please add references.

18. In Line 75, authors say "Polypharmacological drugs may provide a solution to this problem by simultaneously modulating tau aggregation..." This Intro would be strengthened by adding instances when Polypharmacological have been successfully leveraged in clinical therapy.

19. In Line 102 the Authors open by suggesting the delta K280 mutant is linked with AD. This is a liberal perspective taken somewhat out of context even from the literature the authors cite, in which a single case of an AD victim is discussed happening to score positive for the the delta K280 mutant. AD is broadly regarded as spontaneous and not having association with mutated tau. Striking the word AD from this sentence would align the statement better with evidence and the perspective of excerpts in the AD field.

20. Line 116, After focal activation... Do the authors mean "After focal (PHOTO?)activation"? If so, please revise such instances. It is important to be consistent in terminology and precise with language in technical manuscripts.

Response to the reviewers' comments

We would like to thank the reviewers for their thoughtful comments and are pleased that they agree with us that the topic of our study is “*clearly important*” (reviewer 1) and “*of high relevance to neurodegeneration*” (reviewer 2) and that our “*work is of general interest and importance*” (reviewer 3).

Based on the reviewers' comments, we conducted several additional experiments and are confident that we can now address the points raised by the reviewers. Most importantly, we conducted and reported experiments to determine the effect of PHOX15 in primary neurons to bridge the gap “*between the results and developing this work towards a potential therapeutic*” (Reviewer 2). Indeed, our new data now show that the presence of PHOX15 reduces tau aggregation in primary neurons over time. The data is included in Figure 4B. We also performed additional experiments to confirm the specificity of staining with the optotracer Amytracker™ (Reviewer 3). We analyzed the time dependence of Amytracker staining and show that tau increasingly forms amyloids in primary neurons in parallel with the decrease in tau-microtubule interaction (Fig. 2E). We also included a control with non-transduced cells (Fig. 2E), which shows high specificity of Amytracker staining. The first reviewer found the logic of the paper hard to follow because some of the experiments were performed on different tau structures. We optimized the organization of the paper and performed additional molecular dynamics (MD) and post-MD simulations on human 2N4R tau, which is now the same isoform that we used for the cell-free aggregation assays and the cellular assays. The new results are shown in Figure 7D. Since we noticed some reviewer misunderstandings regarding our quantitative live cell imaging assays, we have clarified the description of the experiments, drawing on our previous studies with multiple controls and the mathematical background of the application of our refined reaction-diffusion model (Igaev et al. (2014) *Biophys. J.*; Conze et al. (2022) *Mol. Psychiatry*). We hope that it is now clear how the changes in the effective diffusion constant (D_{eff}) can be used to quantify the interaction of tau with microtubules in axons of living neuronal cells.

Below is a detailed response to the reviewer's comments:

Reviewer #1 (Remarks to the Author):

This paper presents potential inhibitors of tau aggregation, 2-phenyloxazole (PHOX) derivatives. PHOX15 is shown to act by two mechanisms; firstly restoring microtubule binding and inhibiting tau aggregation and secondly inhibiting the tau kinases GSK3 β and Cdk5, reducing tau phosphorylation. It uses a nice fluorescence decay after photoactivation assay to measure microtubule binding in cells. Given the role of tau aggregation in neurodegenerative disease the development of new therapies is clearly important. However, I found the logic of the paper hard to follow. There are cell-free experiments using heparin to induce aggregation, cell experiments and then modelling and confusingly these are all appear to be done on different structures of tau filaments meaning that there are some key questions and additional experiments that need to be addressed before publication to support the conclusions about the mechanisms in the paper:

We streamlined the organization of the paper and focused the experiments on the longest CNS isoform of human tau, which contains 441 amino acids (2N4R tau isoform). We strongly believe that the application of different assay systems (cell-free aggregation assays, cell-free kinase assays, cellular microtubule interaction assays and cellular aggregation assays) are crucial to carefully characterize a polypharmacological drug candidate such as PHOX15 and to analyze the various activities of PHOX15, though this makes the description of the experiments more complex..

1. Tau exists in 6 isoforms. In AD a mixture of 3R and 4R isoforms aggregate while in tauopathies either 3R or 4R tau aggregates. This paper uses wt 2N4R tau and TauAK280. It also mentions fetal tau (3R tau) in the methods although this does not seem to have been used in the experiments? The structure of heparin induced tau filaments has been shown by Goedert and co-workers to be different from those found in human brain. The filaments are also different between AD and 4R tauopathies where just 4R tau form filaments such as in PSP and CBD. This being the case it is hard to understand why all the modelling as I understand it, is done on AD filaments not 4R filaments from a tauopathy and /or the filaments formed with heparin and the relevance of modelling AD protofilaments to the experimental data? It seems much more logical to include or focus on 4R protofilaments in this section to model the cell experiments and possibly also include the modelling the heparin induced filaments. Are the same pockets and glycine triads present in 4R filaments and can PHOX15 bind to these?

To answer this point, we performed additional molecular dynamics (MD) and post-MD simulations on a crystal structure of human 4R tau from Progressive Supranuclear Palsy (PSP) (PDB ID: 7P65), which is the same isoform that we used in the cell-free aggregation assays and the cellular assays. The results are shown in Figure 7D. Together with the previously performed calculations for 4R-3R tau (PDB ID: 5O3L), the newly performed MD-related in silico analyses allowed us to better consider the effects of PHOX15 on early stages of tau aggregation, consistent with our live cell imaging assay performed on 4R tau. The calculations performed allowed the identification of two pockets in 4R-3R tau (i.e. P2 in PDB ID: 5O3L) and 4R tau (i.e. P5 in PDB ID: 7P65), both lined by glycine triads, and presenting shape and binding features that can potentially bind PHOX15. Notably, the results of the in silico analyses suggest that binding of PHOX15 to tau reduces the ability of glycine triads of each filament to adopt a PHF-like conformation in both tau conformations (see Table 1), with this effect is even more pronounced in the 4R structure of tau.

2. The aggregation of tau in the presence of heparin is complicated by heparin acting as an inducer and bypassing the nucleation phase of the aggregation reaction. A ThT kinetics assay in the presence and absence of PhOX 15 may provide additional evidence that it is an inhibitor of aggregation and may elucidate its mechanism to complement the data in Figure 3. If this worked a concentration dependent study could be performed.

We agree that the interpretation of the cell-free aggregation assays is complicated by the presence of heparin. Therefore, we performed additional experiments in cells. We showed that tau amyloid formation in transduced primary neurons increases as a function of expression

time (Figure 2E). We then determined the effect of PHOX15 addition after one week and two weeks of expression of the aggregation-prone tau construct (Figure 4B). We observed significantly fewer AmytrackerTM-positive tau aggregates in PHOX15-treated cells compared to vehicle control at both time points. Note that filaments increasingly formed after one and two weeks in the presence and absence of PHOX15, suggesting that PHOX15 inhibits filament formation but does not dissolve tau amyloids once they have formed in the cell.

3. The effect of PHOX on cells is an interesting result showing that this restores microtubule binding. In figure 1F the authors show that the cells expressing Tau Δ K280 develop aggregates over time. It would be very convincing if they could show that the presence of PHOX15 reduces tau aggregation over time in these cells. Is it possible to include this experiment? Without this result it seems that there is a significant gap between the results and developing this work towards a potential therapeutic which significantly lessens the impact of the paper.

We performed additional experiments to determine the effect of PHOX15 in primary neurons. Indeed, we showed that the presence of PHOX15 reduced tau aggregation in these cells over time (Fig. 4B). We agree that these additional experiments close the gap between the results and the development of this work toward a potential therapeutic.

Reviewer #2 (Remarks to the Author):

The microtubule-associated protein tau is subject to hyperphosphorylation and forms pathological aggregates that contribute to neurodegeneration. In this study, the authors apply a live-cell imaging approach based on photoactivation to monitor pathological changes of human tau regarding its interaction with microtubules. They identify the small molecule PHOX15 to inhibit kinases that are known to phosphorylate tau. A molecular dynamics simulation further suggests that PHOX15 could bind to critical regions within the protofilament and might inhibit initial steps of tau aggregation.

The study addresses an interesting topic of high relevance to neurodegeneration, however requires major revision and is not suitable for publication in its present form. Many data are based on indirect imaging and do not directly control the binding of tau to microtubules. Similar live cell imaging assays based on recovery after photobleaching have been previously used in the field. In any case, results from these assays require biochemical control experiments to independently monitor tau-microtubule interactions. The PHOX15 compound is an interesting molecule but seems to have dual effects on kinase activity and protofilament aggregation. Most of the effects are in the range of 10% or less and require further controls.

We do not believe that the data is based on “indirect imaging”. We developed a quantitative live cell imaging approach based on fluorescence decay after photoactivation (FDAP) experiments to directly measure tau’s interaction in the cell. We have now achieved unprecedented sensitivity in determining changes in the effective diffusion constant (D_{eff}), allowing to directly quantify the interaction of tau with microtubules in axon-like processes of living neuronal cells. In a number of previous studies, we have performed several controls,

such as the use of triple PAGFP, the use of tau constructs lacking the microtubule-binding region, the use of tau phosphomimicking constructs, as well as the use of microtubule-modulating drugs such as epothilones (Weissmann et al., Traffic 2009; Niewidok et al., Mol. Biol. Cell 2016; Conze et al., BRB 2022). Furthermore, we validated the approach with an independent method – single-molecule tracking to directly visualize the movement of individual tau molecules (Janning et al., Mol. Biol. Cell 2014) – and developed a refined reaction-diffusion model by generalizing the standard two-state reaction-diffusion equations (Igaev et al., Biophys. J. 2014). In a very recent publication in “Molecular Psychiatry”, we employed the assay to demonstrate differences in the binding kinetics of caspase-3 cleaved tau to microtubules. Therefore, the assay was extensively validated and crucial controls were carried out. We have added relevant information in the Results section.

Classical biochemical microtubule binding assays do not provide kinetic information and do not reflect the topology of microtubules in axons. They are therefore unable to monitor the effect of PHOX15 on tau-microtubule interaction in cells. In particular, we have shown that the heterogeneity of binding sites in an axon cannot be ignored when it comes to the reaction-diffusion of cytoskeleton-associated proteins (Biophys. J., 2014). To directly determine tau amyloid formation in the cells, we performed additional experiments using an optotracer (Figure 2E). Furthermore, we can now provide direct data that PHOX15 reduces tau amyloid formation in the neurons (Figure 4B), confirming the dual effect of PHOX15 in reducing tau aggregation and decreasing tau phosphorylation.

Specific Points:

1. Title: the term “physiological” regulation is not yet justified. Biochemical experiments to study tau-microtubule interactions are required in neurons.

Following the reviewer’s comment, we changed “physiological regulation” to “physiological interaction” as we agreed that further experiments focusing on the analysis of microtubule dynamics would be required to make a conclusion about microtubule regulation.

2. It needs to be shown that tau forms aggregates (e.g. using oligomer-specific antibodies) and that PHOX15 alters these aggregates.

We performed additional experiments using the optotracer Amytracker™ to demonstrate that tau forms aggregates and to quantify aggregate formation. We observed that tau increasingly forms amyloids over time in primary neurons and have included the data in the manuscript (Figure 2E). We also show that PHOX15 significantly reduces aggregates when added after 1 week and 2 weeks in culture. The new data was included in Figure 4B.

3. The western blot in Figure 1A requires clarification. What is shown in which lane?

We have improved the description in the legend: “Coomassie Brilliant Blue-stained SDS-PAGE of the purified proteins (top left)” is displayed. We also modified the figure to clarify that the two recombinant proteins are shown.

4. Figure 1C: The lower right graph is a quantification of microtubule-bound tau. How was it calculated? The live imaging assay measures diffusion but not tau-MT binding.

Representative images should be added to the graphs.

We have added a schematic representation of the relationship between tau binding and effective diffusion in Figure 1B to clarify that D_{eff} and MT-binding are calculated by fitting the decay transient through a mathematical model to yield mobility and binding properties of the target molecule. We previously developed differential equations for the reaction-diffusion of the tau protein in a cylindrical cellular process, evaluated the full numerical solution, constructed analytical solutions, and validated the approach with single-molecule tracking data and Monte Carlo simulations, as described in detail previously (Igaev et al., Biophys. J. 2014). We have explained the procedure in the text and provided the relevant references (line 114-119).

5. Figure 1F: Amytracker detects tau aggregates. How did the authors quantify the binding of Tau Δ K280 to microtubules based on immunostaining? Representative images for individual time points and quantification should be included.

The binding of Tau Δ K280 to microtubules was determined using the effective diffusion constant (D_{eff}) measured by FDAP experiments, which allows to directly quantify the interaction of tau with microtubules in axon-like processes of living neuronal cells. We have clarified this point in the Figure legend (now Fig. 2F).

6. Figure 2D: the graph quantifies MT bound tau (%). No representative images are shown. Indirect binding/unbinding via diffusion requires an independent binding experiment.

MT-bound tau was determined using the effective diffusion constant (D_{eff}) measured by FDAP experiments. We have clarified this point in the Figure legend (now Fig. 3D).

7. Figure 3A, B: is this result restricted to Tau Δ K280? A second tau mutant should be included. Why is wildtype tau detected in aggregates?

The data show that the activity of PHOX15 to reduce tau aggregation is not limited to Tau Δ K280 but also occurs with wildtype Tau (now Fig. 5B); thus, it is shown that PHOX15 affects tau aggregation independently of the Δ K mutation. We do not see which additional tau mutant should be tested. The experiments are carried out using *in vitro* tau aggregation, where it is shown in Fig. 1A that wildtype tau also forms aggregates with heparin, although to a lesser extent.

8. Figure 3D: Indicate the antibody used for detection. No loading control is shown. Quality of the image is weak.

The PHF1 antibody was used for detection (Greenberg et al., 1992). The information was included in the Methods section and legend (now Fig. 5D). The blot shows the pellet and supernatant fraction, which represents the entire sample. We increased the contrast of the blot.

9. Figure 4: PHOX15 affects GSK3beta, CDK5 (B) and other kinases (E), all of which have multiple substrates. specific is this approach? Are there any side effects following PHOX15

treatment? Control experiments are required to test whether neuronal transmission remains normal

PHOX15 is a polypharmacological drug candidate that affects multiple kinases and results in decreased tau phosphorylation. However, it is also clear that the increased tau phosphorylation seen in Alzheimer's disease and other tauopathies is likely due to multiple kinases and kinase inhibitors are in clinical trials. Further studies need to determine possible side effects. We also agree that future experiments need to analyze the effects of PHOX15 in a systemic context including animal studies, and we have mentioned this in the discussion. However, this goes beyond this study, which aims to identify novel tau aggregation inhibitors that restore physiological microtubule interaction.

10. Figure 4G: effects are in the range of about 5% and I am surprised that they are highly significant based on the number of n-values. An independent complementary approach is required to confirm the FDAP data. The Y-axis should not be interrupted in graphs.

The data show that at any given time point, more than 90% of tau is bound to microtubules, consistent with previous data. A 5% increase in binding seen in the data suggests that the amount of unbound tau has more than halved. Therefore, the effect is quite drastic, but difficult to grasp from the data representation. Since we were calculating the amount of tau bound to microtubules, we decided to stick with this type of plot, although the effect would appear more dramatic if we had plotted the amount of unbound tau. To compensate, we decided to break the Y-axis as the effect is visible between 90 and 100%. As developed above, classical biochemical microtubule binding assays do not reflect the topology of microtubules in axons and are therefore unable to monitor the effect of PHOX15 on tau-microtubule interaction in cells.

11. Page 15, line 456 and Table S5: mention whether PHOX15 does cross the blood-brain barrier.

In silico predictions by using the QikProp software (Table S5) confirmed that PHOX15 is able to cross the blood-brain barrier. This data has been added to the manuscript.

12. Figure 5: This theoretical dataset is very interesting, however no experimental data are included to test this hypothesis. This should be added.

In this manuscript, we used molecular dynamics simulations to investigate possible mechanisms of action of PHOX15 on structural grounds. The simulations are based on the experimental data and the effect of PHOX15 was determined by docking calculations using the IFD (Schrödinger suite 2021-1) into the newly characterized P2 and P5 pockets (Figure 7B and 7D). We also performed a series of MD simulations starting from these tau/PHOX15 complexes, which showed that the compound influences the propensity of the glycine triads to adopt a PHF-like conformation (Figures S20-S22). Future experimental work will specifically address this point.

13. PHOX15 seems to have dual effects on kinases and protofilaments. How can this be explained? Which effect is dominant? Hypo-phosphorylated Tau per se has a lower tendency

to form aggregates. Maybe, the proposed loss in Tau aggregation after application on PHOX15 is only/mainly based on the inhibitory effect of PHOX15 on the Tau kinases.

The entire approach was motivated by the concept of polypharmacology to identify compounds that simultaneously modulate multiple signaling pathways involved in the pathogenesis and progression of complex diseases. The dual effect is explained by PHOX15 having the pharmacophore requirements for binding to both the kinase (hydrogen bonding to the hinge region and adjacent ATP site residues) and tau (see our cited article doi: 10.3390/molecules26165039). This was a major success given PHOX15's relatively low molecular weight and drug-like properties.

Therefore, we searched for possible additional targets of PHOX15 derivatives using an integrated 2D and 3D computational similarity approach. Our combined cell-free and cellular data clearly demonstrate that PHOX15 fulfills this concept by inhibiting tau aggregation (as seen from cell-free data lacking phosphorylation of tau) and inhibiting phosphorylation (as seen from the cell-free data and the phosphoproteomic data). Our data also show that both effects contribute to reducing tau aggregation in cells and restoring microtubule interaction and likely act together. We have provided additional experiments confirming that PHOX15 inhibits tau aggregation in primary neurons.

14. PHOX15 should at least be added to neurons (better: injected to brains) derived from an Alzheimer's disease mouse model. Does it affect pathology in cell culture or in vivo?

In accordance with the reviewer's suggestion, we conduct additional experiments in which we added PHOX15 to primary neurons at one week and two weeks after tau Δ K expression to directly determine its effect on tau amyloids. We then determined the effect of PHOX15 addition of the aggregation-prone tau construct. We observed significantly fewer AmytrackerTM-positive tau aggregates in PHOX15-treated cells compared to vehicle control at both time points. The new data was added as Figure 4B.

We agree that future experiments need to analyze the effect of PHOX15 in a systemic context including animal experiments, and we mentioned this in the discussion. However, this goes beyond this study, which aims to identify novel tau aggregation inhibitors that restore physiological microtubule interaction.

Reviewer #3 (Remarks to the Author):

The manuscript by Pinzi et al establishes live-cell imaging approach to track colocalization of tau bound with microtubules. By using this model and combining with chemoinformatic analyses and molecular dynamics simulations, the authors report discovery of PHOX15, a small molecule that inhibits tau aggregation, restores the physiological microtubule interaction of tau, and inhibits kinases associated with phosphorylating and inducing tau aggregation. The work is of general interest and importance, however, is lacking in crucial detail and experimental controls which need to be addressed before publication.

1. Binding of PHOX15 to tau is inferred from docking and MD studies and suggests PHOX15

interferes with ability of tau to achieve the PHF conformation. But tau folds in many different ways to achieve fibrils with varied structures. How does binding of PHOX15 affect the variety of possible tau fibril folds? This is especially important since it is likely that the experimental models the authors use could adopt alternate tau fibril polymorphs with structures that differ from the ones seen in AD.

As pointed out, tau can adopt a number of different conformations, each of which has a different relevance to tauopathies. To answer this point, we performed molecular dynamics (MD) and post-MD simulations on an additional crystal structure of 4R tau from Progressive Supranuclear Palsy (PDB ID: 7P65); this conformation of the protein was chosen to better assess the effects of PHOX15 on early stages of tau aggregation and also to be consistent with our live cell imaging assay performed on 4R tau. Notably, the calculations performed overall allowed the identification of two pockets into 4R-3R tau (i.e. P2 in PDB ID: 5O3L) and 4R tau (i.e. P5 in PDB ID: 7P65), both lined by a key glycine triad motif and presenting shape and binding features able to accommodate PHOX15; induced fit docking calculations performed on the different tau conformations confirmed good complementarity. In particular, the consequences of PHOX15 binding to these pockets were examined with MD and it was found that the ability of the glycine triads of each filament to adopt a PHF-like conformation is significantly reduced, with this effect even more pronounced in the 4R structure. These results are consistent with the experimental results presented in this work, which indicate that PHOX15 affects the early phases of Tau aggregation. The ability of tau to form channel-like pockets such as those described in this article has never been described before. It could be that these pockets, regardless of the tau conformation from which they originate, have a major influence on the dynamic behavior of tau with respect to its aggregation ability. Of course, further experimental work is required to investigate this dynamic behavior in more detail.

2. In Line 68 the authors say “Mechanisms of toxicity are also controversial, but prefibrillar tau oligomers and soluble tau with disease-like modifications can be toxic”. It is surprising there is no mention here of toxicity by fibrils given that tau inclusions are the observation the authors refer to in Line 60 when justifying tau as a therapeutic target. The Intro would be strengthened by the authors providing some perspective or at least mentioning tau fibrils.

Based on the reviewer's comment, we have changed the text. It now states: “Mechanisms of toxicity are also controversial, but in addition to larger tau fibrils, soluble tau oligomers and tau with disease-like modifications can be toxic species (Fath et al., 2002; Ghag et al., 2018; Patterson et al., 2011).” (lines 67-69)

3. How does the delta K280 mutant affect interaction of tau with microtubules, and how can one extrapolate results of this study to AD, which is vastly a disease of non-mutated tau?

The tau delta K280 mutant, as a tau variant showing increased aggregation, was used to screen for tau aggregation inhibitors. We made this point clear in the introduction (line 104-106). To generalize our results to AD, which is a non-mutated tau disease, we also performed the cell-free assays with wildtype tau (where PHOX15 induces a similar decrease in tau aggregation), tested the effect of PHOX15 on the microtubule interaction of unmutated tau (Fig. 6G, left) and performed the phosphoproteomic analysis on wildtype cells.

4. Loading controls are missing from Fig. 1A.

Figure 1A shows the purified recombinant tau as used for the purity and integrity testing. We have made this point clear in the legend.

5. The interpretation of the authors that higher effective diffusion corresponds with decreased microtubule binding is not well supported. The argument could be strengthened by additional control experiments showing what effects microtubule stabilization and disassembly reagents have on FDAP curves in wt and TauΔK280 cells. These controls are needed to support any interpretation of the tau localization of microtubules.

We have carefully characterized the FDAP assay with multiple controls in several previous publications. There we carried out experiments with triple PAGFP, tau constructs lacking the microtubule binding region, tau phosphomimicking constructs and microtubule-modulating drugs such as epothilones (Weissmann et al., Traffic 2009; Niewidok et al., Mol. Biol. Cell 2016; Conze et al., BRB 2022). In addition, we validated the approach with an independent method - single molecule tracking for direct visualization of tau motility (Janning et al., Mol. Biol. Cell 2014) – and developed a refined reaction-diffusion model by generalizing the standard two-state reaction-diffusion equations (Igaev et al., Biophys. J. 2014). In a very recent publication in “Molecular Psychiatry”, we used the assay to demonstrate differences in the binding kinetics of caspase-3 cleaved tau to microtubules. Therefore, the assay was extensively validated and several controls were carried out. We have added relevant information in the Results section.

6. This reviewer does not follow the logic in Line 127 that increased diffusion indicates reduced tau interaction with microtubules. If tau is nonbound by microtubules, it should have greater diffusion and hence be more difficult to photobleach. Thus, increased decay of TauΔK280 would seem to suggest TauΔK280 is relatively more so bound to microtubules. And wouldn't FRAP be a more appropriate measure of diffusion and microtubule occupancy since tau that is more fractionally bound to microtubules would recover with slower kinetics.

We have clarified the description of the experiment and referred to our previous studies in which we showed that the microtubule interaction reduces the effective diffusion of tau by a factor of 10, as measured by FDAP experiments, and that the change in the effective diffusion constant (D_{eff}) can be used to quantify the interaction of tau with microtubules in living neuronal cells (lines 114-119). We would also like to clarify that the method is based on photoactivation of the tau fusion protein and that the total fluorescence in the cell remains unchanged during the experiments. Thus, the fluorescence decay is only due to the diffusion of the activated population from the activated region (see for example Weissmann et al. Traffic 2009, Figure 1C). Photoactivation is superior to photobleaching because the energy required for activation is much lower than for activation (reducing potential photodamage) and newly synthesized proteins do not contribute to the analysis during the imaging period because they do not fluoresce.

7. To test the hypothesis that puncta corresponding to aggregated (oligomeric?) tau impedes microtubule binding by tau, the authors measure Amytracker staining 3 weeks after

transduction (Line 134). The authors write “aggregates could be observed in most cell bodies”. How is the specificity of the Amytracker confirmed? Control experiments showing the time dependent development of the phenotype post-transduction is needed to confirm correlation of Amytracker signal with tau aggregation, as is Amytracker staining in non-transfected cells.

To address this point, we conducted additional experiments. We analyzed the time dependence of Amytracker staining, which shows an increase with expression time (Fig. 2E), consistent with the decrease in tau-microtubule interaction. We included a control with non-transduced cells (Fig. 2E), which shows high specificity of Amytracker staining and low background fluorescence.

8. Description of the discovery of PHOX15 is lacking in detail. For instance, the authors write “PHOX 5, 8, 15 and 20 were closely related to potent tau aggregation inhibitors”. There are no citations given to link the chemical structures shown in Figure S1 with tau inhibition? Please provide additional citations and supporting information.

At the reviewer's request, additional details regarding the discovery process of PHOX15 were provided. While the ChEMBL ID is usually considered a reference for linking compounds to their in vitro activities, we also added the required citation to PubChem bioassay data from which the activity annotations were derived.

9. In Figure 2E, what is the right most data point labeled Tau PHOX15? What concentration PHOX15 is it? How does it fit into this figure with comparison to Tau delta K280?

The rightmost data point represents the result from wild-type tau with the highest PHOX15 concentration used (25 μ M). To differentiate, we changed the color of this point to dark turquoise. The idea is to present this as the endpoint of full restoration of physiological microtubule interaction.

10. The statement in Line 219: “Cylindrical filaments with periodically thin regions are considered to be the PHF precursor” is unclear but seems to draw correlation between recombinant fibrils and PHFs from AD. This is a dangerous correlation to draw given recent studies that show striking differences between recombinant and AD tau fibrils. Without confirmatory data, equating recombinant fibrils and AD PHFs should be avoided.

Following the reviewer's comment, we rewrote the paragraph and added a note of caution with an appropriate reference: “However, it should be noted that recombinant heparin-induced tau fibrils and tau filaments isolated from patients with tauopathies have striking structural differences (Zhang et al., 2019)” (lines 254-259).

11. Error bars are missing from Fig. 4b

Error bars are missing in Figure 4b because the compound was tested in singlicate. 10 point concentrations were tested, resulting in a curve with very clear sigmoidal shape, indicating a fairly high level of confidence. For consistency, this procedure was adopted in line with our previous testing protocol.

12. *With regard to kinase activity, the authors write: “This binding mode is consistent with the structure-activity relationships (SAR) of other PHOX derivatives tested in this work (data not shown).” What is meant by this? Did experimental screening with other analogs suggest importance for the amide carbonyl, para-hydroxyl, and phenyl rings? If so, these data are crucial validations and should most certainly be shown.*

Based on the reviewer's comment, we have removed this unfortunately misleading sentence. The structure-activity relationships referred to the general requirements for ATP-competitive inhibitors of protein kinases, which were then translated into the PHOX series through chemical similarity arguments. Our biological testing focused only on the best candidate.

13. *Description of molecular docking is absent from the Results section and is inadequately described. The authors write “The pockets were not present in the initial cryo-EM structure 5O3L.” Also the authors write “the structure of the protofilament remained relatively stable during the MD.” How then were the proposed sites of binding deduced if MD failed to open the proposed sites of binding and the sites are not exposed in the beginning structure used?*

We are sorry we didn't make this point clear enough. The channel-like pockets were not present in the original structure but opened during MD thanks to the dynamic behavior of tau. Therefore, docking of PHOX15 into representative conformations of tau was performed with an open channel sampled during MD. Subsequently, MD was also performed on the tau-PHOX15 complexes, starting with the docked complexes. The text has been revised to make this description clearer.

14. *The authors suggest ligand binding might occur due to channels that open in the tau PHF due to conformational and abundance of conformationally flexible glycine residues. This is not supported by the CryoEM data. Although the PDB file 5o3L used for MD was low resolution, which prohibited measures of conformational flexibility, a later structure 6HRE provides experimental B factors. Residues in P2, namely GLY 333-335 have B factors of 42-46. These are quite low compared with other regions of the PHF where B factors sometimes approach 100. Therefore there is general disagreement of the interpretation of high structural fluctuations at sites with Glycines and pre-existing experimental data.*

B factors provide useful clues about protein flexibility, but definite statements based on B factors alone are questionable for intrinsically disordered proteins such as tau. In fact, MD simulations can be very useful in this regard. We showed that both dimeric and monomeric tau were able to open a channel-like pocket near the glycine triad, water molecules flowed through the channel, and the pocket was able to bind PHOX15. Of course, the flexibility was even higher for monomeric tau, which, as discussed, is less structured than dimeric tau.

15. *It seems unlikely that any PHOX analogs enter cavities in the PHF. This is even alluded to by the authors who write that there is no effect seen with incubation of PHOX with PHFs or recombinant fibrils. A more likely mechanism that is proposed by the authors is PHOX binds tau monomers preventing PHF folds from emerging since interaction of PHOX in the cavities would prevent folding of tau into the PHF conformation.*

The pocket opens in both monomeric and dimeric tau. In both cases, we showed that PHOX15 can bind into these cavities and alter the propensity of tau to form PHF aggregates. However, this effect was more evident for monomeric tau, which is consistent with the experimental results presented in this paper and the reviewer's feeling. However, such an effect was observed in both structures and we thought it appropriate to describe it as we see it. It is currently too early to draw any final conclusions. Hopefully more definitive answers will be available in the next few years.

Minor Issues

16. The phrasing: "quantitatively monitor pathological changes of human tau as it interacts with axonal microtubules" in the Abstract reads in a confusing way. Human tau as it interacts with axonal microtubules is a physiological interaction. Dissociation from microtubules might be considered pathologic. Please try to revise the sentence to more clearly articulate these distinctions for the reader.

We rewrote the sentence and it now reads: "We developed a live-cell imaging approach to quantitatively monitor pathological changes of human tau with respect to its interaction with axonal microtubules." (lines 31-34)

17. In Line 73, it is said "Because abnormal forms of tau could trigger a plethora of pathomechanisms". Please add references.

We have added two references at lines 73 and 74 (Chang, C.W., E. Shao, and L. Mucke. 2021. Tau: Enabler of diverse brain disorders and target of rapidly evolving therapeutic strategies. *Science*. 371; Khan, S.S., and G.S. Bloom. 2016. Tau: The Center of a Signaling Nexus in Alzheimer's Disease. *Front Neurosci*. 10:31.)

18. In Line 75, authors say "Polypharmacological drugs may provide a solution to this problem by simultaneously modulating tau aggregation..." This Intro would be strengthened by adding instances when Polypharmacological have been successfully leveraged in clinical therapy.

There are several polypharmacological drugs available in the market that offer higher efficacy, lower drug resistance, and several other potential advantages. We have added an explanatory sentence with references to the introduction (lines 76-78).

19. In Line 102 the Authors open by suggesting the delta K280 mutant is linked with AD. This is a liberal perspective taken somewhat out of context even from the literature the authors cite, in which a single case of an AD victim is discussed happening to score positive for the the delta K280 mutant. AD is broadly regarded as spontaneous and not having association with mutated tau. Striking the word AD from this sentence would align the statement better with evidence and the perspective of excerpts in the AD field.

We rewrote the sentence. It now states: "We used the single amino acid deletion mutant Tau Δ K280 reported in two cases of tauopathies (Momeni et al., 2009; Rizzu et al., 1999) as a tool to develop a cellular assay to test for compounds affecting tau aggregation and improving the tau-microtubule interaction." (lines 104-106)

20. Line 116, After focal activation... Do the authors mean “After focal (PHOTO?)activation”? If so, please revise such instances. It is important to be consistent in terminology and precise with language in technical manuscripts.

We have revised the terminology according to the reviewer's comment to make it more consistent and precise.

REVIEWERS' COMMENTS

Reviewer #2 (Remarks to the Author):

The revised manuscript by Pinzi et al. has been nicely developed. The authors have addressed most of my concerns. In particular, the new figures 2e and 4b significantly strengthen the authors conclusions. A final concern is the selected western blot in Figure 1a. Although the results are quantified over several independent experiments, the selected blot is not fully representative, as a stronger aggregation of deltaK280 is hardly visible in this example. If the authors could exchange this image with a better image this would be appreciated, however this point is not limiting. Together, I can now recommend publication of the project in Nature Communications.

Reviewer #3 (Remarks to the Author):

The revised manuscript by Pinzi et al adds nice controls to live-cell imaging experiments, which improves the overall reliability of the reported effects of PHOX15 on counteracting tau aggregation and increasing its binding to MTs. The FDAP work is compelling and sound. Although, the work suffers in the proposed mechanism of action, which remains highly speculative. Binding of PHOX15 to tau remains conjecture based on docking and MD that is without direct confirmatory experimental evidence. IFD using Schrödinger software is reliable, but ought to be scrutinized with more experimental data if authors wish to validate the supposed binding site. Notably, experimental evidence presented for binding of PHOX15 to GSK3 β and Cdk5 seems sufficient based on data shown in Figure 6. But the claim that ligands bind in cryptic channel-like pockets in tau based on molecular dynamics (MD) simulations is conjecture without experimental validation. This aspect detracts from an otherwise outstanding body of experimental work showing that PHOX15 improves residence time of tau deltaK280 on the MT, and possibly has some effect on counteracting tau aggregation.

A minor, although significant note, is that the PDB structure for Progressive Supranuclear Palsy tau is a cryo-EM, not crystal. This should be corrected regardless of where the paper is published.

Response to the reviewers' comments

We are pleased that the reviewers appreciated the additional experiments we conducted after the first round of reviews. And we are pleased that the reviewers state that “The revised manuscript by Pinzi et al. has been nicely developed“ (reviewer 2) and that „the FDAP work is compelling and sound“ (reviewer 3).

Reviewer 3 noted that the proposed mechanism of action remains highly speculative. We agree that additional work is warranted to confirm the prediction from the molecular dynamics simulations, particularly with regard to the binding site of PHOX15 to the channel-like pockets. However, we believe that this is beyond the scope of this already very extensive study, in which we developed an approach to monitor pathological changes of aggregation-prone human tau in living neurons, investigated 2-phenyloxazole (PHOX) derivatives as putative polypharmacological small molecules, identified a candidate (PHOX15) that inhibits tau aggregation and modulates tau kinases, performed a phosphoproteomic analysis to characterize the effect of the drug on tau phosphorylation in cells, and proposed a possible mechanism of action by molecular dynamics simulation how PHOX15 prevents tau aggregation.

Below is a detailed response to the reviewer's comments:

Reviewer #2 (Remarks to the Author):

A final concern is the selected western blot in Figure 1a. Although the results are quantified over several independent experiments, the selected blot is not fully representative, as a stronger aggregation of deltaK280 is hardly visible in this example. If the authors could exchange this image with a better image this would be appreciated, however this point is not limiting:

We suspect that there is a misunderstanding on the part of the reviewer. Figure 1a shows Coomassie Brilliant Blue stained SDS-PAGE of the purified proteins used for the aggregation assays (no Western blot for tau aggregation) to demonstrate the purity of the recombinant protein preparation. We have made this point clear in the figure and the legend.

Reviewer #3 (Remarks to the Author):

A minor, although significant note, is that the PDB structure for Progressive Supranuclear Palsy tau is a cryo-EM, not crystal.

Thank you for pointing out this error. We have corrected this in the manuscript and supplementary material.